# Effects of high-intensity interval training versus moderate-intensity continuous training on cardiorespiratory and exercise capacity in patients with coronary artery disease: A systematic review and meta-analysis

Chao Gao[1], Yuchuan Yue[2]\*, Dongmei Wu[2]\*, Junming Zhang[1], Shuyao Zhu[1]

**1** Chengdu University of Traditional Chinese Medicine, School of Nursing Chengdu, Chengdu, Sichuan, China, **2** Chengdu Fourth People's Hospital, Chengdu, Sichuan, China

\* 1903222804@qq.com (YY); wudongmei_2001@163.com (DW)

## Abstract

### Background

With the increasing utilization of cardiac rehabilitation in clinical treatment and prognosis for patients with cardiovascular diseases, exercise training has become a crucial component. High-intensity interval training (HIIT) and moderate-intensity continuous training (MICT) are commonly employed in rehabilitating patients with cardiovascular diseases. However, further investigation is required to determine whether HIIT and MICT can effectively enhance the prognosis of patients with coronary artery disease. Therefore, this study aims to assess the effectiveness of HIIT and MICT interventions, optimal intervention duration for different intensity levels of training, as well as effective training modalities that improve cardiorespiratory function and exercise capacity among patients.

### Methods

We conducted a comprehensive search of the Cochrane Library, PubMed, EMbase, Web of Science, and CINAHL databases for randomized controlled trials (RCTs) pertaining to high-intensity interval training (HIIT) and moderate-intensity continuous training (MICT) interventions in patients with coronary artery disease from inception until publication on September 26, 2024. Two independent researchers assessed articles that met the inclusion criteria and analyzed the results using Sata 17.0 software. Forest plots were employed to evaluate the impact of HIIT and MICT on outcome indicators. Sensitivity analysis and funnel plot assessment were performed to examine publication bias. Subgroup analysis was conducted to determine optimal intervention duration and training methods.

### Results

A total of 22 studies with 1364 patients were included in the study, including the HIIT group (n = 685) and the MICT group (n = 679). The results showed that compared to MICT, HIIT

**Data Availability Statement:** All relevant data are within the paper and its Supporting Information files.

**Funding:** Project number: XJLL2023007; Funded by: The Fourth People's Hospital of Chengdu; Sponsor: Dongmei Wu(Writing – original draft); Yuchuan Yue (Preparation of the manuscript).

**Competing interests:** The authors have declared that no competing interests exist.

significantly increased PeakVO$_2$(Peak oxygen uptake)[WMD = 1.42mL /kg/min 95%CI (0.87, 1.98), P = 0.870, I$^2$ = 0%], 6MWT(6-minute walk test)[WMD = 18.60m 95%CI (2.29, 34.92), P = 0.789, I$^2$ = 0%], PHR(Peak heart rate)[WMD = 4.21bpm 95%CI (1.07, 7.36), P = 0.865, I$^2$ = 0%], DBP(diastolic blood pressure)[WMD = 3.43mmHg 95%CI (1.09, 5.76), P = 0.004, I$^2$ = 60.2%]. However, in LVEF(left- ventricular ejection fraction)[WMD = 0.32mL 95%CI (-1.83, 2.46), P = 0.699, I$^2$ = 0%], LVEDV(left ventricular end-diastolic volume)[WMD = 0.91 ml 95%CI (-1.83, 2.46), P = 0.995, I$^2$ = 0%] and SBP(systolic blood pressure)[WMD = 1.85mmHg 95%CI (-0.23, 3.93),P = 0.266, I$^2$ = 18.2%], there was no significant difference between HIIT and MICT.

## Conclusion

Based on the findings of this systematic review, HIIT demonstrates superior efficacy compared to MICT in enhancing PeakVO$_2$, PHR, 6MWT and DBP. However, no significant differences were observed in LVEF, LVEDV, and SBP. In summary, HIIT exhibits potential for improving cardiopulmonary function and exercise capacity among patients with coronary artery disease.

## 1 Introduction

The incidence and mortality of cardio vascular diseases (CVD) have been steadily increasing over the years [1]. According to a study on the global burden of cardiovascular diseases and risk factors, the total incidence of cardiovascular diseases has risen from 271 million in 1990 to 523 million in 2019. Among these cases, coronary artery disease (CAD) stands out as one of the leading causes of global mortality [2]. The prevalence of CAD continues to grow, inevitably resulting in an escalation of global healthcare costs and economic burdens [2]. With advancements in clinical medicine, percutaneous coronary intervention (PCI) and coronary artery bypass grafting (CABG) have emerged as effective treatment methods for CAD patients, facilitating blood perfusion restoration and improvement in clinical symptoms [3]. However, adverse cardiovascular events still manifest among certain individuals with CAD following PCI. Enhancing prognosis and rehabilitation outcomes for CAD patients has thus become a pivotal clinical concern [4].

With the implementation of comprehensive management strategies for patients with CVD, cardiac rehabilitation (CR) has gained international recognition as a Class 1A recommendation for enhancing exercise performance and prognosis [5]. It has demonstrated positive and beneficial effects on individuals with acute coronary syndrome (ACS) and CAD [6]. Exercise training is an integral component of CR, which has been shown in relevant studies to improve cardiorespiratory function, ventricular filling, vascular endothelial function, and reduce mortality and morbidity among CAD patients [7,8]. Guidelines for exercise in CVD patients recommend utilizing aerobic exercises at varying intensities [9,10], including high-intensity intermittent exercise (HIIT), which involves short bursts of high-intensity activity interspersed with recovery periods. Compared to moderate intensity continuous training (MICT), HIIT exerts more favorable effects on cardiopulmonary, peripheral, and metabolic systems. However, HIIT is typically suitable for stable CVD patients due to its demanding nature in terms of activity intensity and cardiopulmonary requirements. Conversely, MICT is often preferred by patients due to its steady exercise intensity and moderate demands on maximum active heart rate [9].

Currently, there is no consensus on the optimal exercise mode for CAD patients between HIIT and MICT. As a result, more RCTs and systematic reviews are utilizing both HIIT and MICT to intervene in the cardiopulmonary health and quality of life of cardiovascular disease patients [11–13]. However, previous systematic reviews have encountered issues such as increased heterogeneity due to inclusion of heart failure and coronary heart disease patients, suboptimal exercise intervention duration leading to weakened patient compliance, inconsistencies in outcome indicators, as well as differences in optimal intervention time, frequency and duration of each exercise. Therefore, this study aims to conduct a meta-analysis using PeakVO$_2$ as the primary outcome indicator to evaluate the effects of HIIT and MICT on exercise capacity and cardiopulmonary health among CVD patients. The results will provide actionable recommendations for improving cardiopulmonary function through identifying the best exercise mode for these individuals.

## 2 Methods

### 2.1 Protocol and registration

The Preferred Reporting Items for Systematic Review and Meta-Analyses (PRISMA) guidelines were followed for the methodology of this review (S1 File). The complete protocol of this meta-analysis was uploaded and registered on the PROSPERO platform with the registration number:CRD42024532872.

### 2.2 Search strategy

The relevant literatures meeting the criteria in PubMed, Cochrance Library, EMbase, Web of Science and CINAHL databases were searched by computer, and the search period was from the establishment of the database to Sep 26, 2024. Search requires a combination of subject words and free words. The English search terms were "Coronary Artery Disease", "Myocardial Ischemia", "Acute Coronary Syndrome", "percutaneous coronary" intervention, Myocardial Infarction, and high intensity interval training were followed up on references from relevant systematic reviews or meta-analyses. See attachment (S2 File) for specific search methods.

### 2.3 Study selection

Inclusion criteria: (1) Study type: randomized controlled trial. (2) Study population: patients with CAD (Coronary artery disease), including those who have undergone PCI(percutaneous coronary intervention) and CABG(Coronary Artery Bypass Grafting), regardless of gender, duration of the disease, or age; Patients with ischemic heart disease and myocardial infarction meeting the diagnostic criteria outlined in the guidelines[14]. (3) Interventions: A comparative study comparing high-intensity intermittent exercise (HIIT) and moderate-intensity sustained exercise (MICT) for a minimum of 4 weeks. (4) Outcome indicators: At least one of the following outcomes was measured: peak oxygen uptake (PeakVO$_2$), 6-minute walking test (6MWT), maximum heart rate (PHR), left ventricular ejection fraction (LVEF), left ventricular end-diastolic volume(LVEDV), systolic blood pressure(SBP), and diastolic blood pressure(DBP).

Exclusion criteria: (1) Non-randomized controlled trials. (2) Incomplete data. (3) Studies involving patients with other severe comorbidities.

### 2.4 Data extraction

Use EndNoteX21 for document management. The literature was independently screened by two investigators (Gao/Zhang) who cross-checked the following information: (1) Basic study information, including first author, publication time, country, and study type; (2) Baseline

characteristics of the study population such as age, sex, sample size, and disease type; interventions including exercise type, duration, and frequency; (3) Key elements of biased risk assessment; (4) Result indicators such as PeakVO$_2$, 6MWT, PHR etc. Mean and standard deviation changes were calculated according to the Cochrane Manual [15] using baseline and endpoint mean values along with their respective standard deviations. Any discrepancies were resolved through discussion or negotiation with third parties. Initially, articles were screened based on titles followed by further evaluation of abstracts and full texts to determine inclusion eligibility. Missing data was obtained by contacting original study authors via email or phone.

## 2.5 Methodological quality assessment

Two investigators independently assessed the included RCTs for bias risk using the Cochrane Manual 5.0.1 criteria [15]. The assessment covered random sequence generation, allocation concealment, blinding methods, data integrity, selective outcome reporting, and other potential sources of bias. Each dimension was categorized as "yes" (indicating low risk of bias), "no" (indicating high risk of bias), or "unclear" (indicating medium risk of bias). Studies demonstrating low risk of bias across all dimensions were considered to have an overall low risk of bias; studies with any dimension rated as high risk were classified as having an overall high risk of bias. In cases where there was disagreement between the two reviewers' assessments, a third reviewer would be consulted.

## 2.6 Quality of evidence

The Recommendation and Evaluation Grading (GRADE) utilizes a web-based version (https://gradepro.org) for assessing the quality of evidence. Based on the GRADE standard for grading evidence quality, this study divided the evaluation of literature quality and criteria for downgrading into five items: risk of bias, risk of inconsistency, risk of indirectness, risk of imprecision, and other factors. The level of their risks was evaluated accordingly [16]. Differences in quality evaluation are resolved through discussions among researchers, and if no consensus can be reached, senior researchers are consulted.

## 2.7 Statistical analysis

The meta-analysis was conducted using Stata 17.0 software. All the data extracted in this study are continuous variables. If the outcome indicators were the same, weighted mean difference (WMD) was used for effect size comparison. If the outcome indicators were different, standardized mean difference (SMD) was used for effect size comparison. The I$^2$ test was utilized to assess heterogeneity, with an I$^2$ of 25–50% indicating low heterogeneity, 50–75% indicating moderate heterogeneity, and >75% indicating severe heterogeneity. When I$^2 \leq$50%, a fixed-effect model is employed; when I$^2 >$50%, a random-effects model is adopted. The results of the meta-analysis were presented in forest plots format. Funnel plot and Egger's test were employed to evaluate publication bias detected by a simple graphical method [17]. Sensitivity analysis and subgroup analysis were performed to examine the source of heterogeneity, while intervention time was divided into three subgroups ($\leq$6 weeks, 8–12 weeks,$\geq$12weeks). The exercise methods were categorized into two sub-groups: (treadmill and cycle ergometer).

## 3 Results

### 3.1 Study selection

A total of 1432 articles were searched, and after repeated checks, 1112 were retained. Subsequently, the articles underwent pre-screening by reading the title and abstract. Among them,

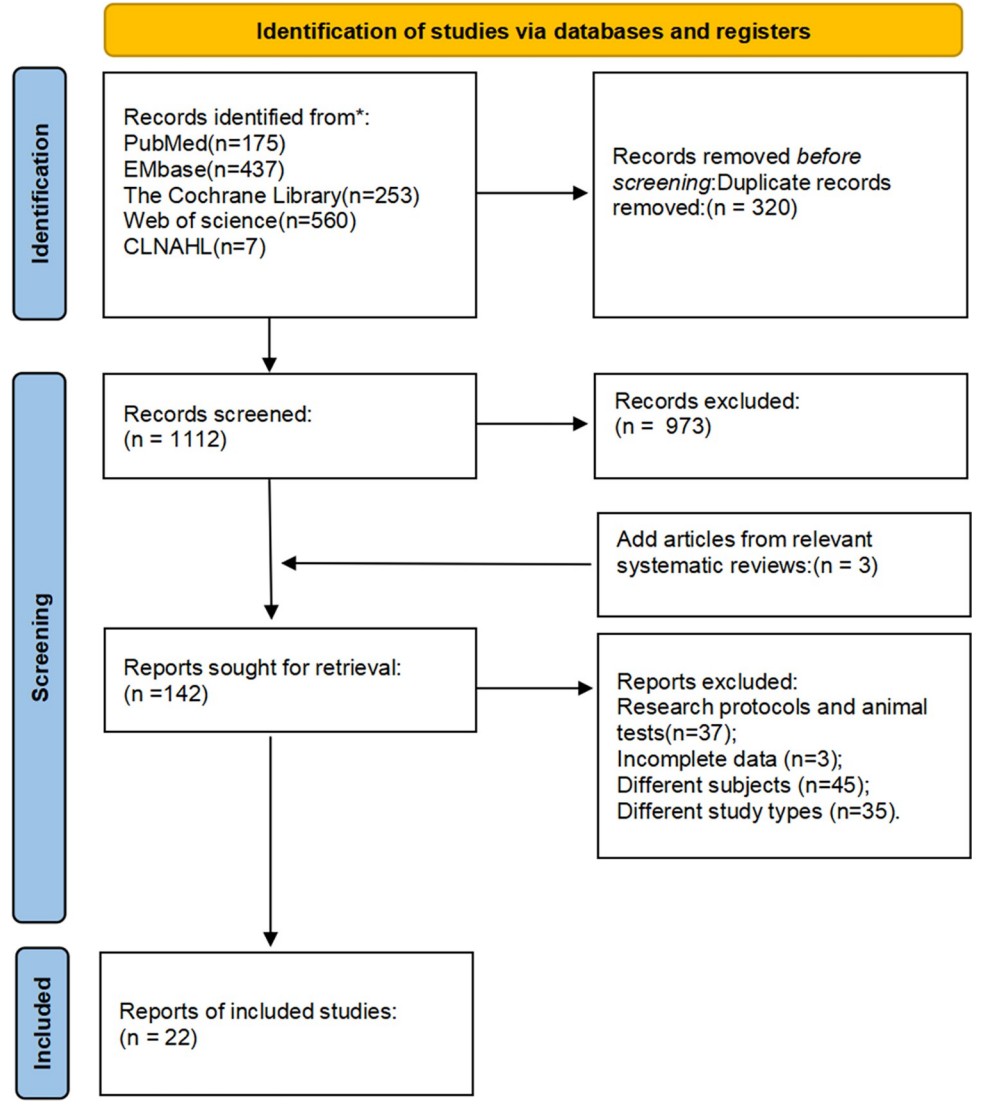

**Fig 1. Flowchart of selection of included studies.**

142 were evaluated for full-text reading, out of which 45 did not match the subjects; 35 had different study types; 3 had incomplete data; 18 were related to program meetings and studies; and 19 involved animal experiments. Finally, a total of 22 articles met the inclusion criteria. The flow chart for document screening is shown as Fig 1.

## 3.2 Study characteristics

The 22 randomized controlled studies included in this analysis were conducted in 13 countries, namely Canada, the United States, Brazil, the United Kingdom, Switzerland, Spain, Portugal, Australia, Turkey, Egypt, Iran, South Korea and other countries. The majority of participants had coronary artery diseases such as PCI, GABG, and myocardial infarction. The exercise interventions ranged from 4 to 16 weeks in duration with each exercise session lasting between 28 minutes and 40 minutes. Tables 1 and 2 present the key characteristics of the included literature.

**Table 1. Basic features of the studies.**

| ID | Study | Year | country | partcipant | Simple(T/C) | Sex(M/F) | Age(T/C) | Outcome | Setting |
|---|---|---|---|---|---|---|---|---|---|
| 1 | Ha-Yoon Choi | 2018 | Korea | MI | 23/21 | 39/5 | T:53.00±6.84 C:57.31±12.62 | ①② | rehabilitation clinic |
| 2 | Abdelhalem | 2018 | Egypt | CAD | 20/20 | 34/6 | T:54.65±7.63 C:51.95±8.07 | ④ | Ain Shams University Hospital |
| 3 | Aispuru-Lanche | 2023 | Spain | MI | 28/28 | 47/9 | T:58.9 ±8.0 C:58.9 ±8.0 | ④⑤ | rehabilitation clinic |
| 4 | Cardozo | 2015 | Brazil | CAD | 24/23 | 31/16 | T:56 ± 12 C:62 ± 12 | ①③⑥⑦ | rehabilitation clinic |
| 5 | Currie | 2015 | Canada | MI, PCI, CABG | 11/9 | 18/1 | T/C 63 ± 8 | ①③⑥⑦ | Hamilton Health Sciences General Site |
| 6 | Currie | 2013 | Canada | MI, PCI, CABG | 11/11 | 20/2 | T: 62 ± 11 C: 68 ± 8 | ①③⑥⑦ | Hamilton Health Sciences General Site |
| 7 | Dunford | 2021 | Canada | MI, PCI, CABG | 9/9 | 16/2 | T:62 ± 8 C:62 ± 6 | ①③⑥⑦ | Hamilton General Hospital |
| 8 | Eser | 2022 | Switzerland | PCI | 35/34 | 69 | T:56 ± 10 C:59 ± 10 | ①④⑤⑥⑦ | Bern University Hospital |
| 9 | Ghardashi-Afousi | 2018 | Iran | CABG | 14/14 | 28 | T:53.90±3.44 C:54.10±4.02 | ③④⑤⑥⑦ | Baqiyatallah hospital |
| 10 | Gonçalves | 2023 | Portugal | CAD | 23/23 | 36/12 | T:50 ± 9 C:55 ± 10 | ① | Espírito Santo Hospital |
| 11 | Jaureguizar | 2016 | Spain | MI, CAD | 36/36 | 61/11 | T/C:58 ± 11 | ①②③⑥⑦ | \ |
| 12 | McGregor | 2023 | England | MI, PCI, CABG | 187/195 | 356/26 | T:58.9±9.2 C:59±9.9 | ①⑥⑦ | \ |
| 13 | Reed | 2022 | Canada | PCI, CABG | 43/44 | 74/13 | T: 61 ± 7 C: 60 ± 7 | ②⑥⑦ | University of Ottawa Heart Institute |
| 14 | Taylor | 2020 | Australian | CAD | 47/46 | 78/15 | T:61 ± 7 C:61 ± 8 | ②③⑥⑦ | rehabilitation clinic |
| 15 | Terada | 2022 | Canada | CAD | 31/30 | 48/13 | T/C:61 ± 7 | ②⑥⑦ | rehabilitation clinic |
| 16 | Jaureguizar | 2019 | Spain | MI, CAD | 57/53 | 92/18 | T:57.6 ± 9.8 C:58.3 ± 9.5 | ①③ | |
| 17 | Yakut | 2022 | Turkey | MI | 11/10 | 18/3 | T:59.6 ± 4.5 C: 58.5 ± 5.6 | ②⑥⑦ | Dokuz Eylül University Hospital |
| 18 | Okur | 2022 | Turkey | CABG | 7/7 | \ | T:59.14± 3.63 C:62.00± 6.61 | ①② | Kütahya Health Sciences University Hospital |
| 19 | Kim | 2015 | Korea | MI | 14/14 | 22/6 | T:57±11.58 C:60.2±13.64 | ①③ | Sanggye Paik Hospital |
| 20 | Trachsel | 2019 | Canada | MI | 9/10 | 6/13 | T: 60 ± 10 C:57 ± 13 | ①③⑥⑦ | Montreal Heart Institute |
| 21 | Keteyian | 2014 | America | MI, PCI, CABG | 15/13 | 23/5 | T:60 ± 7 C: 58 ± 9 | ①③⑥⑦ | Henry Ford Hospital |
| 22 | Nam | 2023 | Korea | PCI | 30/29 | 52/7 | T:56.07±10.48 C:58.69±12.38 | ①②③ | \ |

T:interventiongroup;C:controlgroup;M:Male;F:Female;MI:myocardial infarction;PCL:percutaneous coronary intervention;CABG:Coro-nary Artery Bypass Grafting;①PeakVO2(Peak oxygen uptake);②6MWT(6-minute walk test);③PHR(Peak heart rate);④LVEF(left- ventricular ejection fraction);⑤LVEDV(left ventricular end-diastolic volume);⑥SBP(systolic blood pressure);⑦DBP(diastolic blood pressure).

### 3.3 Quality assessment

Two investigators (Gao/Zhang) used the Cochrane Review Manual to conduct a rigorous quality evaluation of the included literature. Among the 22 [18–30] studies included, 9 [24,25,28–34] used random number tables or computers to generate random numbers, and 13 [18–23,26,27,35–39] studies mentioned randomness without specifying the specific method.8

**Table 2. Basic features of the studies.**

| ID | Study | Additional stages | Interventions | Model | Intensity | Duration | Frequency | Time |
|---|---|---|---|---|---|---|---|---|
| 1 | Ha-Yoon Choi | 10minWU,5minCD | H:high intensity sprint for 4 min, active recovery for 3 min, alternating 4 sets | / | 85%-100%HRmax alternating 50%-60% HRmax | 10W | 1~2/w | 28min |
| | | \ | M:moderate intensity exercise all the time | | 60%-80% HRmax | | | |
| 2 | Abdelhalem | 5minWU,5minCD | H:high intensity sprint for 2–5 min | Treadmil | 85%-95%HRmax | 12W | 1/w | 35min |
| | | \ | M:moderate intensity running all the time | | 40%-60% HRmax | | | |
| 3 | Aispuru-Lanche | \ | H:high intensity running for 10min alternating active recovery for 4 min | Treadmil | 85%-95%HRmax | 16w | 2/w | 20min |
| | | \ | M:Moderate intensity exercise less than 10 minutes, active recovery for 4 minutes | | 65%-85%HRmax | | | |
| 4 | Cardozo | 5minWU,5minCD | H:high intensity sprint for 4 min,active recovery for 2 min | Treadmil | 60%-90%HRmax | 16w | 3/w | 30min |
| | | 5minWU,5minCD | M:moderate intensity cycling all the time | | 70%-75%HRmax | | | |
| 5 | Currie | 10minWU,10minCD | H:high intensity sprint for 1 min, alternating 4 sets | Cycle ergometer | 1-4w80%-104%PPO at start, (increased by 10% every 4 weeks) 5-8w:102%PPO;9-12w:110%PPO | 12W | 2/w | 30min |
| | | 10minWU,10minCD | M:moderate intensity cycling all the time | | 51%-65%PPO | | | |
| 6 | Currie | 10minWU,10minCD | H:high intensity sprint for 1 min, alternating 11 sets | Cycle ergometer | 1-4w80%-104%PPO at start, (increased by 10% every 4 weeks) 5-8w:102%PPO;9-12w:110%PPO | 12W | | 30min |
| | | 10minWU,10minCD | M:moderate intensity cycling all the time(11minCD) | | 51%-65%PPO | | | |
| 7 | Dunford | \ | H:High intensity 6 steps up or down the stairs, active recovery 90s, alternating 3 sets | \ | 75%-90%HRmax | 8W | 3/w | 30min |
| | | \ | M: moderate intensity exercise all the time | | 60%-80%HRmax | | | |
| 8 | Eser | \ | H:high intensity sprint for 4 min, active recovery for 3 min, alternating 4 sets | Cycle ergometer | 75%-90%HRmax | 8W | 3/w | M:30min |
| | | 10minWU,3minCD | M: moderate intensity cycling all the time | | 60%-80%HRmax | | | |
| 9 | Ghardashi-Afousi | \ | H:high intensity cycling for 2 min, moderate intensity cycling for 2 min, active recovery for 10s,alternating 10 sets | Treadmil | 85%-95%HRmax alternating 50%HRmax | 6W | 3/w | 40min |
| | | 5minWU | M:moderate intensity running all the time | | 70%HRmax | | | |
| 10 | Gonçalves | 10minWU,5minCD | H:high intensity sprint for 4 min, active recovery for 1 min, alternating 4 sets | Treadmil | 85%–95%HRpeak; Active recovery 40%HRpeak | 6W | 3/w | H:30m M:28min |
| | | \ | M:moderate intensity running all the time(10minWU,5minCD) | | 70%-75%HRpeak | | | |

*(Continued)*

**Table 2.** (Continued)

| ID | Study | Additional stages | Interventions | Model | Intensity | Duration | Frequency | Time |
|---|---|---|---|---|---|---|---|---|
| 11 | Jaureguizar | 5-10minWU,5-13minCD | H:high intensity sprint for 20s active recovery for 40s | Cycle ergometer | The first month: 20s(50%Workload);40s actively recover 10%Workload. The second month: 20s(50%Workload) alternately40s (10%Workload); 40s Active recovery of 10% | 8W | 3/w | 40min |
| | | 5-10minWU,5-13minCD | M:moderate intensity cycling all the time | | 62%-75%HRmax | | | |
| 12 | McGregor | \ | H:high intensity sprint for 1 min, alternating 10 sets | Cycle ergometer | 85%-95%PPO;active recovery20%-25%PPO | 8W | 1/w | 20min |
| | | \ | M:moderate intensity cycling all the time | | 40%-70%HRmax | | | |
| 13 | Reed | 10minCD | H:high intensity sprint for 4 min, active recovery for 3 min, alternating 4 sets | Treadmil | 85%-95%HRmax alternating 60%-70%HRmax | 12W | 2/w | H:45min M:60min |
| | | \ | M:moderate intensity running all the time | | \ | | | |
| 14 | Taylor | \ | H:high intensity sprint for 4 min, active recovery for 3 min, alternating 4 sets | Treadmil | 85%-95%HRmax | 4W | 3/w | H:32min M:40min |
| | | \ | M:moderate intensity running all the time | | 65%-75%HRmax | | | |
| 15 | Terada | \ | H:high intensity sprint for 4 min, active recovery for 3 min, alternating 4 sets | Cycle ergometer | 85%-95%HRmax alternating 60%-70%HRmax | 12W | 2/w | H:45min M:60min |
| | | \ | M:moderate intensity running all the time | | Exercise at an intensity of 20 or 40 per minute above normal heart rate | | | |
| 16 | Jaureguizar | 10minWU,13minCD | H:high intensity sprint for 20s active recovery for 40s | Cycle ergometer | The first month: 20s(50%Workload);40s actively recover 10%Workload. The second month: 20s(50%Workload) alternately40s (10%Workload); 40s Active recovery of 10% | 8W | 3/w | 40min |
| | | 10minWU,13minCD | M:moderate intensity cycling all the time | | 62%-75%HRmax | | | |
| 17 | Yakut | 10minCD | HIIT:high intensity sprint for 4 min, active recovery for 3min, moderate intensity cycling for 3min,alternating 4 sets | Stair climbing | 85–95%HRmax alternating 70%HRmax | 12W | 2/w | H:28min M:20-45min |
| | | 10minCD | M:moderate intensity exercise all the time | | 70%-75%HRmax | | | |
| 18 | Okur | 10minCD | H:high intensity sprint for 4 min, active recovery for 3 min, alternating 4 sets | Cycle ergometer | 80%-90%Wmax alternating 50%-70%Wmax | 5W | 5/w | H:28min M:30-40min |
| | | 10minCD | M:moderate intensity cycling all the time | | 50%-70%Wmax | | | |
| 19 | Kim | 10MinCD | H:high intensity sprint for 4 min, active recovery for 3 min, alternating 4 sets | Treadmil | 85%-95%HRmax alternating 50%-70%HRmax | 6W | 3/w | 45min |
| | | 10MinCD | M:moderate intensity running all the time | | 70%-85%HRmax | | | |

(*Continued*)

**Table 2.** (Continued)

| ID | Study | Additional stages | Interventions | Model | Intensity | Duration | Frequency | Time |
|---|---|---|---|---|---|---|---|---|
| 20 | Trachsel | 5minCD | H:high intensity sprint for 6–8 min, active recovery for 5 min, alternating 2-3sets | Cycle ergometer | 85%-95%HRmax | 12W | 5/w | H:24min M:30-60min |
| | | 5minCD | M:moderate intensity cycling all the time | | 70%-75%HRmax | | | |
| 21 | Steven J Keteyian | 5minWU,4minCD | H:high intensity sprint for 4 min, active recovery for 3 min, alternating 4 sets | Treadmil | 80%-90%HRmax alternating 60–70%HRmax | 12W | 2/w | H:28min M:30min |
| | | 5minWU,4minCD | M:moderate intensity running all the time | | 60%-80%HRmax | | | |
| 22 | Hoon Nam | 10MinCD | H:high intensity sprint for 4 min, active recovery for 3 min | Treadmil | 95%-100%$VO^2_{max}$ alternating 60%$VO^2_{max}$ | 9W | 2/w | 28min |
| | | 10MinCD | M:moderate intensity for 2 min,active recovery for 3 min | | 80%$VO^2_{max}$alternating 60% $VO^2$ | | | |

WU:Warm up;CD:Calm down;H:HIIT;M:MICT;HRmax:maximal heart rate;HRpeak:Peak heart rate;PPO:peak power output;$VO^2_{max}$:maximal oxygen consumption.

[24,25,28–33] studies focused on allocation hiding, with 4 [25,29,30,32] explaining the use of opaque envelopes for allocation hiding. 7 [28–33,38] studies implemented blinding, including 2 [28,38] that blinded evaluators and 5 [29–33] that blinded patients. 18 [18–26,29,30,32–38] studies mentioned the sites of sports intervention, with 13 [19,20,22–26,29,30,33,34,36,38] indicating specific intervention sites. The evaluation indicators and results are shown in Fig 2.

### 3.4 GRADE of evidence

According to the summary of evidence from the grading Recommendations and Assessment Development and Evaluation (GRADE), moderate quality was found for $PeakVO_2$ and PHR, while very low quality was found for 6MWT, LVEF, LVEDV, SBP, and DBP. The reasons for this degradation may include: (1) most studies did not specify the use of allocation hiding or blind methods; (2) sample sizes were insufficient for three outcome indicators - 6MWT, LVEF, and LVEDV; (3) confidence intervals were too wide for three outcome indicators— LVEF, LVEDV, and SBP; (4) funnel plots showed asymmetry in four outcome indexes - 6MWT, LVEF, SBP,and DBP (GRADE of evidence S3 File).

### 3.5 Results of meta-analysis

**3.5.1 $PeakVO_2$(Peak oxygen uptake).**   All 16 included studies reported PeakVO2, and the effect of HIIT on the peak value of PeakVO2 was better than that of MICT [WMD = 1.45 mL/kg/min 95% CI (0.90, 2.01), P = 0.908, I2 = 0%] Fig 3.

**3.5.2 6MWT(6-minute walk test).**   Among the 7 included studies, 6MWT was reported. The effect of HIIT on 6MWT was significantly better than that of MICT [WMD = 18.60m 95% CI (2.29, 34.92), P = 0.789, $I^2$ = 0%] Fig 4.

**3.5.3 PHR(Peak heart rate).**   The 11 included studies reported the change in PHR during exercise, and the results indicated that HIIT had a better effect on PHR compared to MICT [WMD = 4.21 bpm 95% CI (1.07, 7.36), P = 0.865, I2 = 0%] Fig 5.

**3.5.4 Left ventricular function and remodelling.**   The 4 included studies reported the changes of LVEF before and after exercise, and the results showed that there was no significant difference in the improvement effect of HIIT and MICT on LVEF (Fig 6) [WMD = 0.32mL 95%CI (-1.83, 2.46), P = 0.699, $I^2$ = 0%]. The 3 included studies reported the changes of LVEDV before and after exercise, and found that there was no significant difference in the

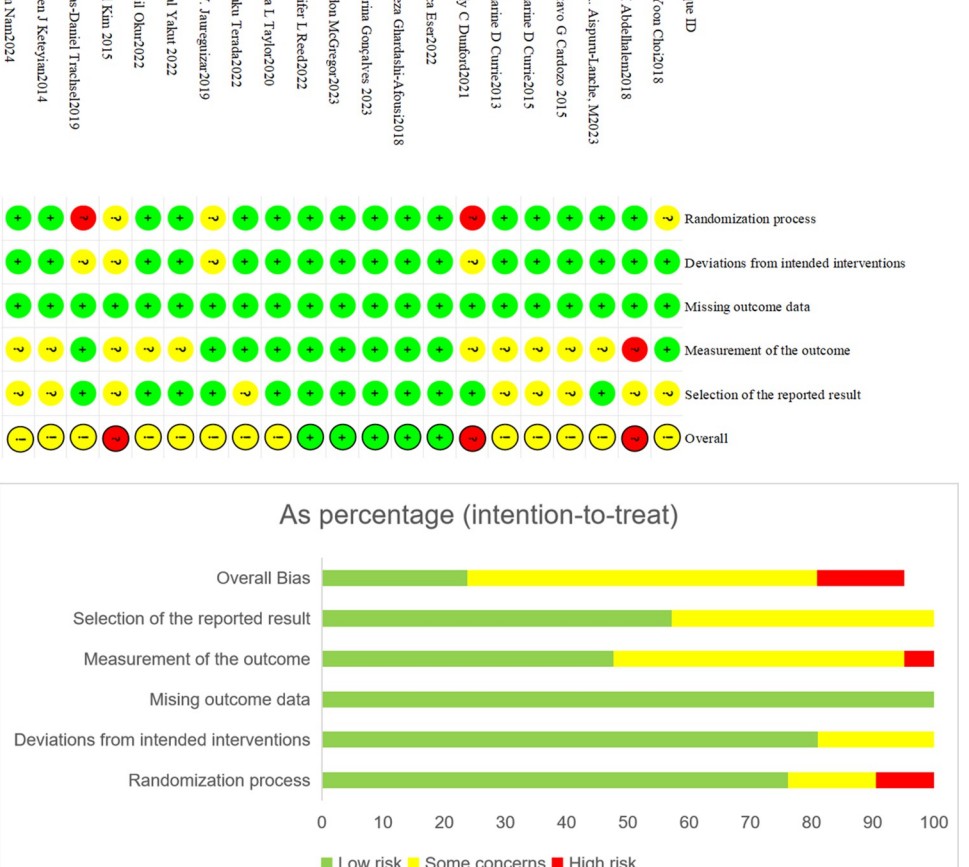

**Fig 2. Risk of bias summary.**

improvement effect of HIIT and MICT on LVEF (Fig 7) [WMD = 0.91mL 95%CI (-3.68, 5.49), P = 0.995, $I^2$ = 0%].

**3.5.5 SBP(systolic blood pressure), DBP(diastolic blood pressure).** The 12 studies included in the analysis reported changes in bp(blood pressure) before and after exercise. The results indicated that HIIT had a significantly greater effect on DBP than MICT [WMD = 3.43mmHg 95%CI (1.09, 5.76), P = 0.004, $I^2$ = 60.2%], (Fig 8) with moderate heterogeneity observed among the studies. However, there was no significant difference between HIIT and MICT on SBP [WMD = 1.85mmHg 95% CI (-0.23,3.93), P = 0.266,$I^2$ = 18.2%] (Fig 9).

## 3.6 Subgroup analysis

The subgroup analysis was conducted based on the intervention duration, which was categorized into three subgroups (≤6 weeks, 8–12 weeks, ≥12 weeks). The measurement of the main outcome index, PeakVO$_2$, after HIIT and MICT interventions revealed that all three intervention durations led to improvements in PeakVO$_2$ [WMD = 1.42 mL/kg/min 95%CI(0.87, 1.98) P = 0.868; $I^2$ = 0%]. Specifically, an intervention duration of ≥12 weeks resulted [WMD = 2.31 mL/kg/min 95%CI(0.55, 4.07),P = 0.919; $I^2$ = 0%] in a greater improvement in PeakVO$_2$ compared to an intervention duration of 8–10 weeks[WMD = 1.35mL/kg/min 95% CI (0.56, 2.14), P = 0.440, $I^2$ = 0%] and ≤6 weeks [WMD = 1.29 mL/kg/min 95%CI(0.42, 2.17), P = 0.731; $I^2$ = 0%] subgroups (Fig 10). According to the protocols of HIIT and MICT, exercise modes were

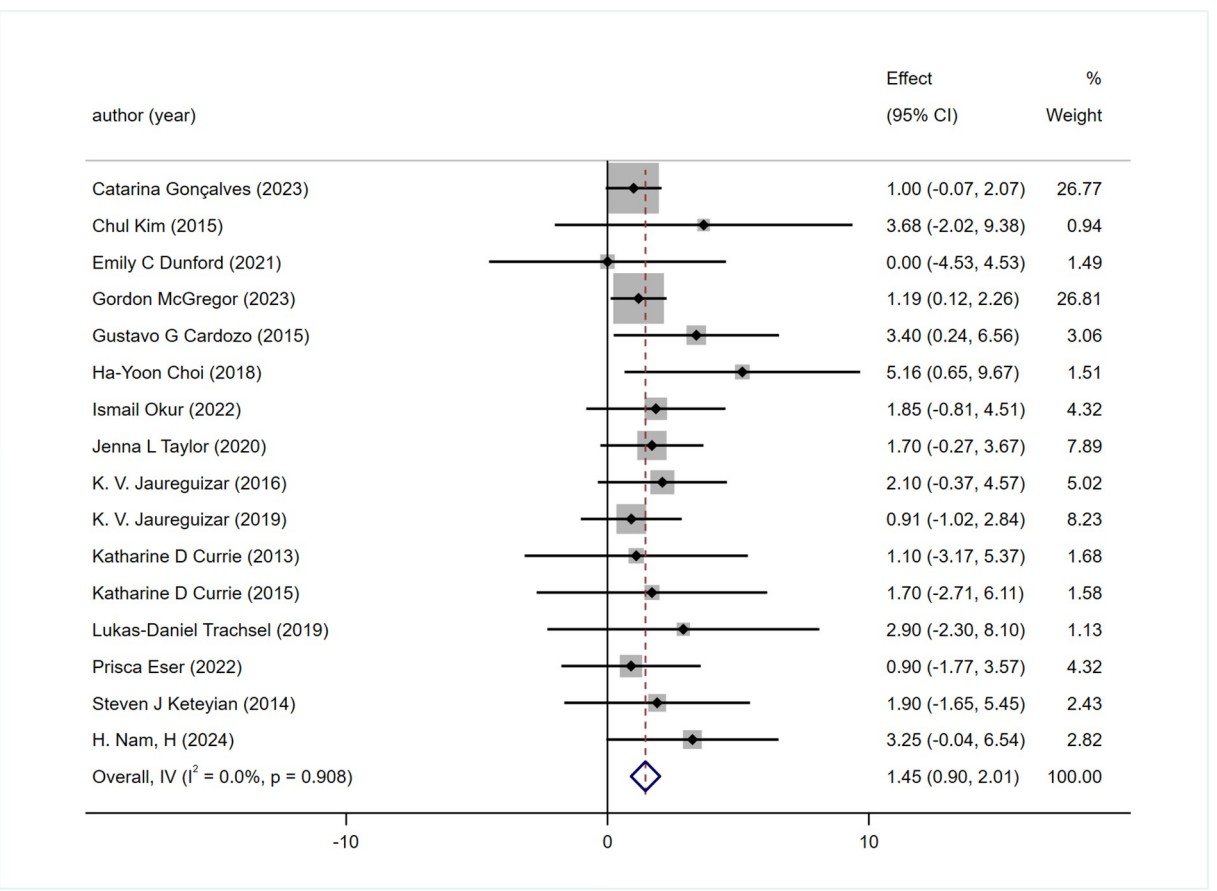

**Fig 3. Forest plot comparing the improvement of peak VO2 between two exercise intensity.**

categorized into two subgroups, namely treadmill and cycle ergometer. The primary outcome measure, PeakVO$_2$, was assessed for changes. Subgroup analysis revealed that both exercise modes demonstrated significant improvements in PeakVO$_2$ [WMD = 1.39 mL/kg/min 95%CI (0.82, 1.95),P = 0.942,I$^2$ = 0%]. Specifically, HIIT and MICT utilizing a treadmill showed a greater effect on improving PeakVO$_2$ [WMD = 1.55 mL/kg/min 95% CI (0.71, 2.38),P = 0.557, I$^2$ = 0%], compared to using a cycle ergometer [WMD = 1.26 mL/kg/min 95%CI (0.49, 2.02), P = 0 .964,I$^2$ = 0%] (Fig 11). According to the weekly frequency change of HIIT and MICT exercises, participants were divided into three subgroups (≤2 times/week, 3 times/week, > 3 times/week), and the change in the main outcome indicator PeakVO$_2$ was measured. The results demonstrated that all three exercise frequencies led to an increase in PeakVO$_2$ [WMD = 1.45mL/kg/min 95%CI (0.90, 2.01), P = 0.908, I$^2$ = 0%]. Specifically, exercising more than three times per week [WMD = 2.07mL/kg/min 95%CI (-0.30, 4.44), P = 0.725, I$^2$ = 0%] yielded better results compared to exercising less than twice a week [WMD = 1.58mL/kg/min 95%CI (0.66, 2.49),P = 0.549,I$^2$ = 0%] or thrice a week [WMD = 1.32 mL/kg/mi n95%CI(0.59, 2.04), P = 0.804,I$^2$ = 0%] (Fig 12).

According to the duration of each exercise session for HIIT and MICT, participants were categorized into three subgroups based on exercise time (< 30 minutes, 30–40 minutes, > 40 minutes), and the change in PeakVO$_2$ was measured. The results from the subgroups indicated that all three durations of exercise led to improvements in PeakVO$_2$ [WMD = 1.45 mL/kg/min 95%CI(0.90, 2.01),P = 0.908,I$^2$ = 0%]. Notably, exercising for more than 40 minutes had a

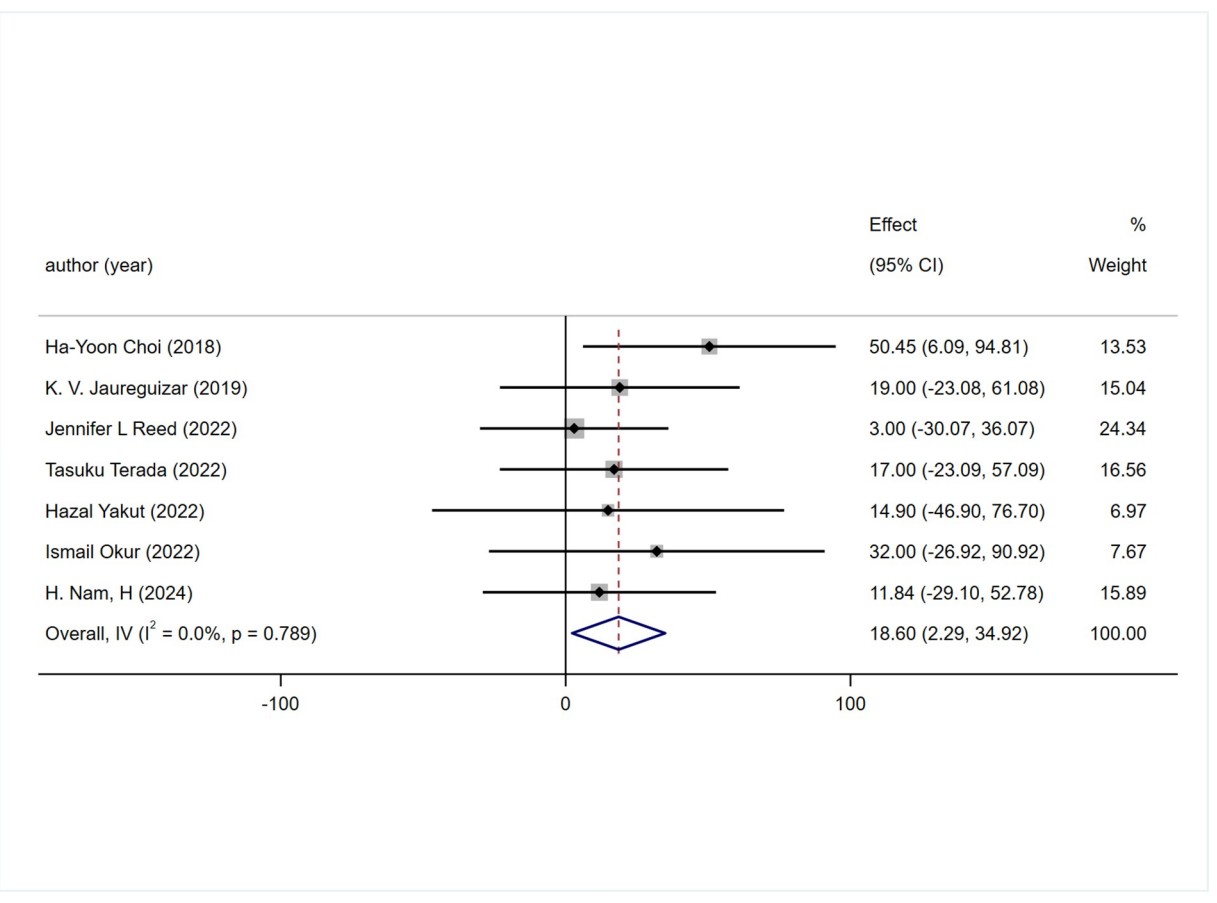

**Fig 4. Forest plot comparing the improvement of 6MWT between two exercise intensity.**

greater impact [WMD = 2.31 mL/kg/min 95%CI(0.12, 4.50),P = 0.824,$I^2$ = 0%] compared to exercising for a duration of 30-40minutes[WMD = 1.51mL/kg/min 95%CI(0.55, 2.47), P = 0.914,$I^2$ = 0%] or lessthan30minutes[WMD = 1.33mL/kg/min 95%CI(0.62, 2.04), P = 0.330,$I^2$ = 13.2%] (Fig 13). However, no significant differences were observed among the aforementioned analysis results regarding intervention duration (P = 0.578), exercise mode (P = 0.616), exercise frequency (P = 0.794), andexercise session(P = 0.701).

## 3.7 Publication bias

The funnel plot and Egger test for the three outcome indicators of PeakVO$_2$, PHR, and BP (SBP and DBP) of this study were drawn using Stata17. The results of the funnel plot showed that the PHR results were distributed symmetrically along both sides of the symmetry axis, with most data falling within the funnel plot indicating a low risk of publication bias.However, the funnel plots for PeakVO$_2$ and BP (SBP, DBP) were asymmetrically distributed, suggesting a potential risk of publication bias. Furthermore, the Egger test revealed no evidence of publication bias in other outcome indicators except for PeakVO$_2$ (t = 2.83,P = 0.031). This may be attributed to variations in intervention content between studies on HIIT and MICT as well as differences in baseline characteristics and intervention effects on PeakVO$_2$ such as male-to-female ratio. In accordance with inclusion criteria and considerations regarding publication bias testing, 6MWT, LVEF, and LVEDV were not included in this analysis due to limited availability (<10 studies)(Egger test diagram S4 File)(funnel plot S4 File).

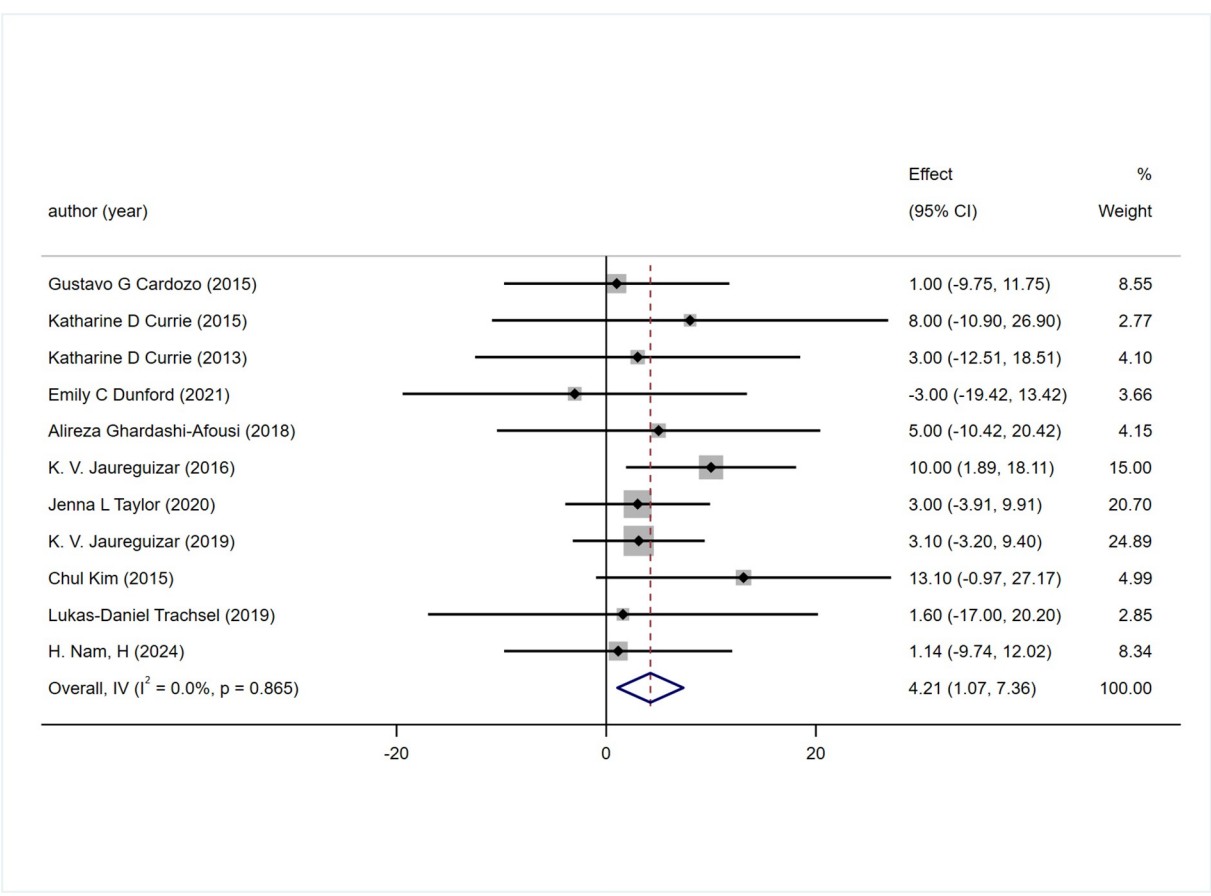

**Fig 5. Forest plot of the effects of high-intensity and moderate-intensity exercise on PHR in patients.**

## 3.8 Sensitivity analysis

Sensitivity analysis was performed on three outcome indicators of PeakVO$_2$, PHR, and BP (SBP, DBP) in this study to assess the impact of each individual study on the overall results (Figs 14–17). After excluding one study at a time, it was found that the total effect size of PeakVO$_2$ from the 16 included studies fell within the original total effect size's 95% CI range, indicating relatively stable sensitivity analysis results. The PHR of 11 studies and BP (SBP, DBP) of 12 studies included in this research remained stable even after excluding two specific studies [27,28]. This could be attributed to Jaureguizar [27] studies completing a higher workload than MICT after finishing HIIT program and having a larger difference in PHR before and after both exercise modes compared to other studies. However, due to its large sample size (HIIT:187/MICT:195), the McGregor [28] study had a significant impact weight on the overall results. Nevertheless, this study's findings revealed no significant difference in blood pressure impact between completing HIIT and MICT programs.

## 4 Discussion

In this systematic review, we included 22 randomized controlled trials involving a total of 1364 patients and found that HIIT had better effects on improving PeakVO$_2$, 6MWT, and PHR than MICT in patients with coronary artery disease. However, there was no significant difference in LVEF, LVEDV, and SBP.

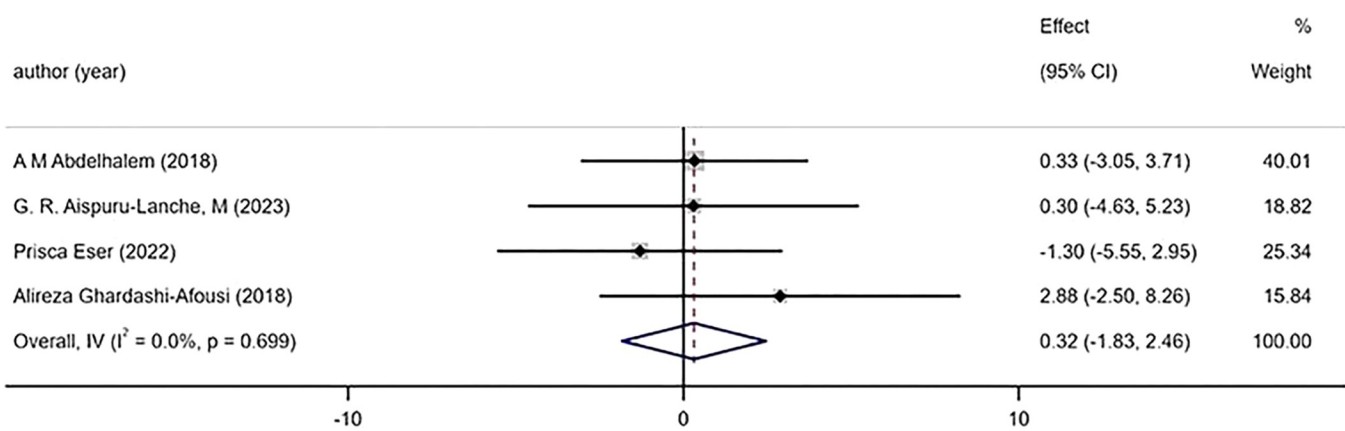

**Fig 6. Forest plot of the effects of high-intensity and moderate-intensity exercise on LVEF in patients.**

Cardiac rehabilitation (CR) is a complex intervention that may involve various therapies, such as exercise, education on risk factors, behavior modification, and psychological support [40]. CR plays a crucial role in contemporary care for patients with cardiovascular disease and is recommended by the European Society of Cardiology, the American Heart Association, and the American College of Cardiology for post-cardiovascular event cardiac rehabilitation. Additionally, exercise therapy constitutes an essential component of cardiac rehabilitation [41]. According to the guidelines, MICT, as a conventional exercise modality in cardiac rehabilitation [42], is well-tolerated by patients with various cardiovascular diseases undergoing treatment or rehabilitation and promotes cardiopulmonary health [43]. However, due to its characteristics of high-intensity exercise within a short duration and subsequent rapid recovery, HIIT can more effectively enhance cardiopulmonary fitness and achieve higher overall exercise intensity, thereby increasing physiological stimulation and significantly improving maximum aerobic capacity [43]. Studies have demonstrated that PeakVO$_2$ serves as an independent predictor of all-cause mortality and specific mortality related to cardiovascular diseases [43] and has been recognized as a vital sign by AHA [44]. Furthermore, exercise intensity during physical activity also plays a crucial role in cardiac protection, with high-intensity exercise inducing notable changes in PeakVO$_2$. Therefore, HIIT can optimize oxygen uptake, transportation, and utilization during exercise to provide substantial stimulus for enhancing the alteration of PeakVO$_2$ [45].

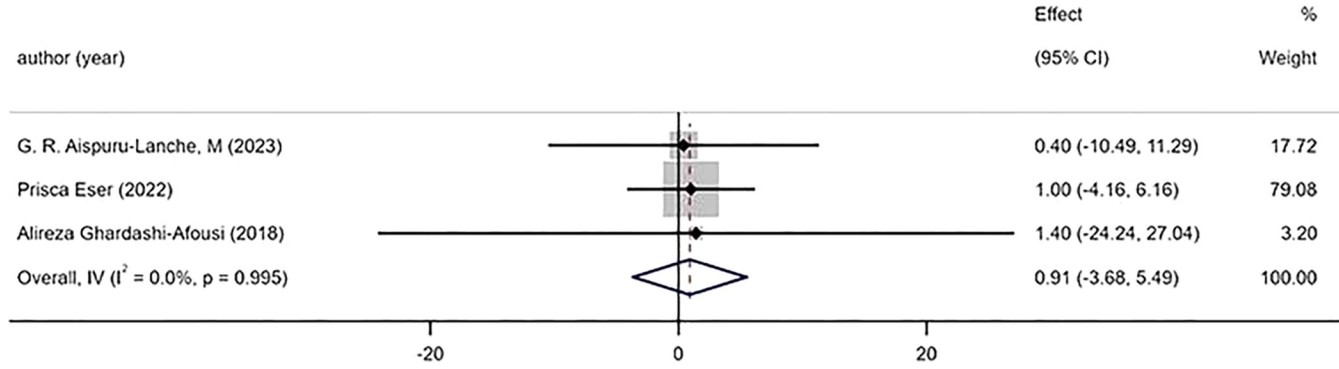

**Fig 7. Forest plot of the effects of high-intensity and moderate-intensity exercise on LVEDV in patients.**

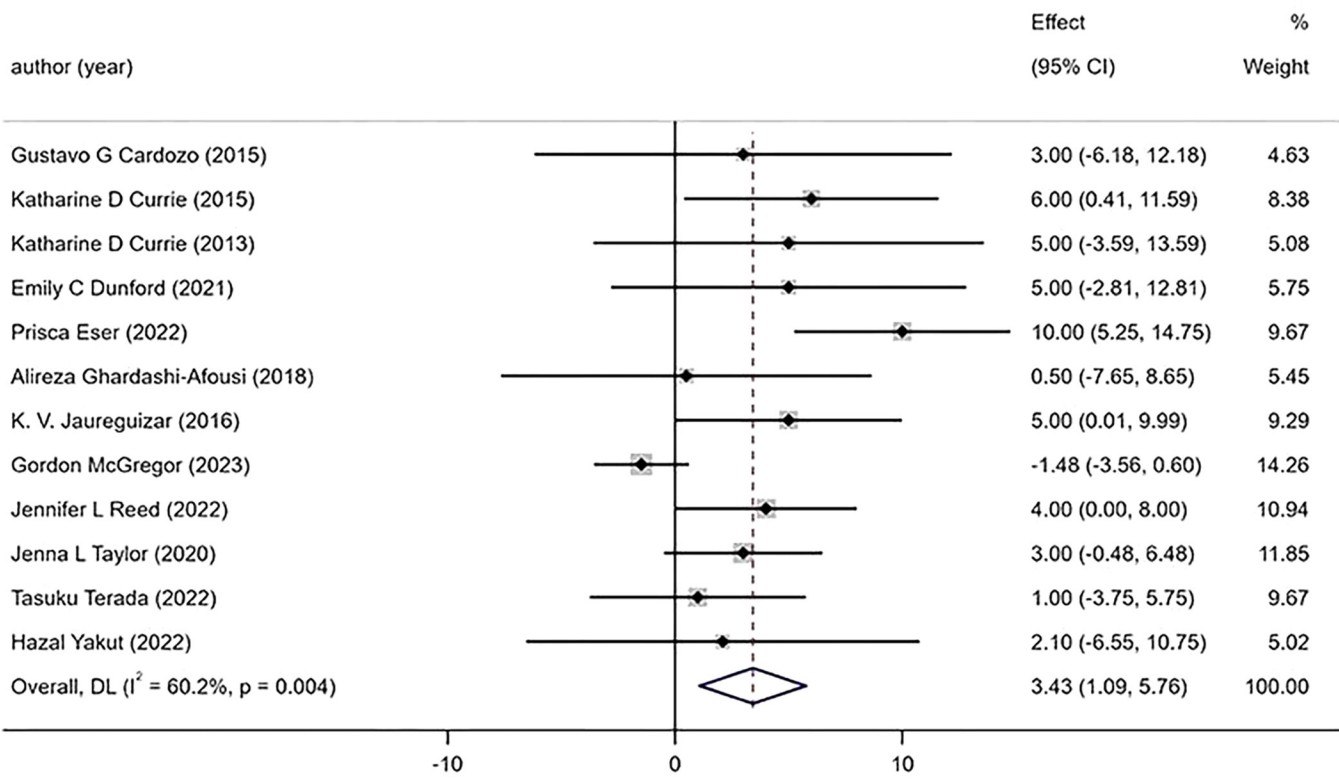

**Fig 8. Forest plot of the effects of high-intensity and moderate-intensity exercise on SBP in patients.**

The findings of this study demonstrated that patients receiving HIIT exhibited a significant increase in PeakVO$_2$ by 1.42mL/kg/min compared to those undergoing MICT, which is consistent with the results reported in recent systematic reviews [11,46]. Therefore, our results provide support for the utilization of HIIT as an exercise regimen to enhance cardiopulmonary function and exercise capacity among individuals with coronary artery disease. Subgroup analysis revealed that intervention durations ≥12 weeks yielded the most substantial improvement in PeakVO$_2$, aligning with the conclusions drawn by Wang et al. [11] and Zheng et al. [5] in their respective systematic reviews. These outcomes differ from those presented by Li et al. [46] (< 6 weeks) and Goncalves et al. [47] (< 12 weeks) in their systematic reviews due to potential heterogeneity within Li's subgroup analysis (which included only two studies < 6 weeks) and low compliance observed after supervised stages of different intensity training. according to Goncalves' review[47] (wherein only one study received both HIIT and MICT supervised courses). Consequently, a relatively favorable intervention effect was observed during the initial six-week period. In another systematic review conducted by Goncalves etal. [47], along with meta-regression analyses aiming to determine optimal training intensity and duration for CVD patients, it was found that moderate-to-vigorous or vigorous exercise can enhance cardiopulmonary fitness, with an ideal training program lasting between 6–12 weeks. However, Pattyn et al.'s findings [48] indicated no significant difference between training durations < 12 weeks and ≥12 weeks regarding improvement effects on PeakVO$_2$ –consistent with our subgroup results obtained from this study. The subgroup analysis of different intensity training modes showed that the intervention effect of treadmill with different intensity

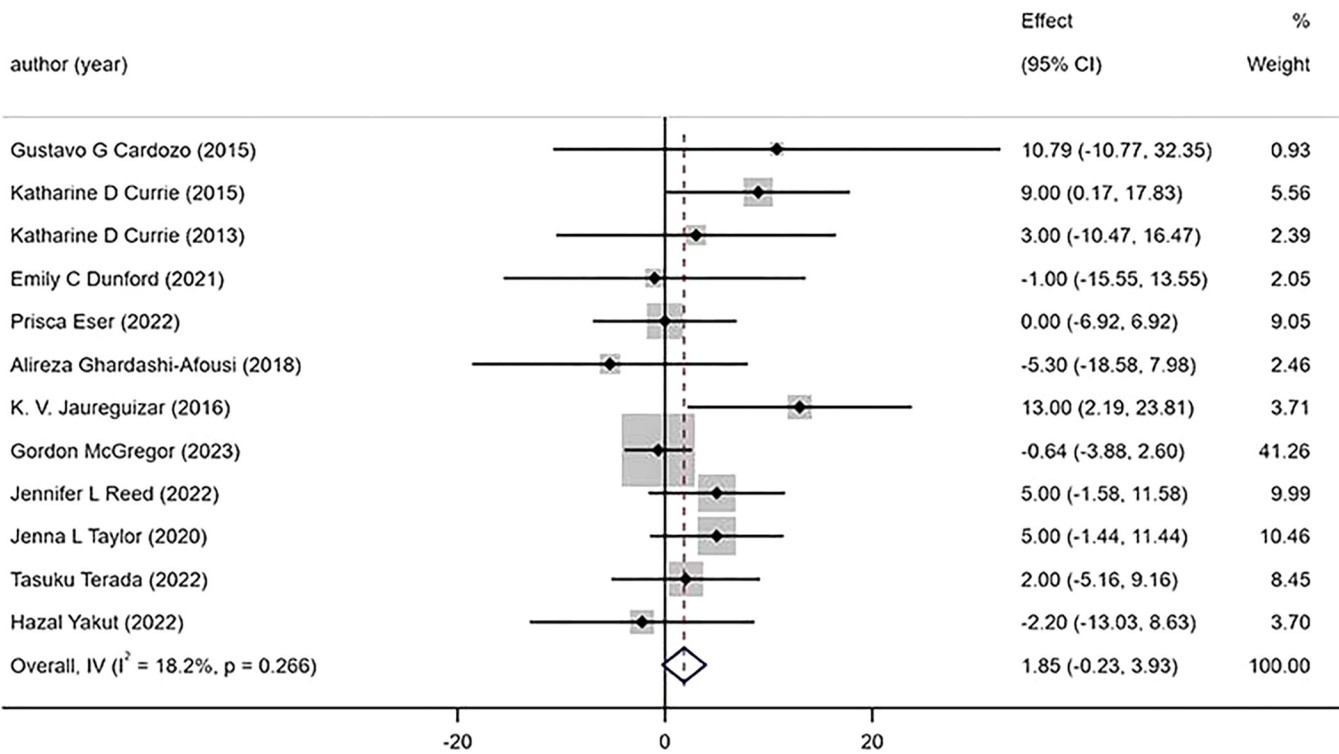

**Fig 9. Forest plot of the effects of high-intensity and moderate-intensity exercise on DBP in patients.**

exercise may be numerically superior to that of a cycle ergometer. However, Du et al.'s study [45] divided HIIT and MICT exercise patterns into three subgroups (treadmill, cycle ergometer, other exercise patterns), and no significant difference was found in the subgroup analysis ($P = 0.75, I^2 = 0\%$), which is consistent with the results of this study. The results of the subgroups showed that treadmill or cycle ergometer at different intensities affected the changes in $PeakVO_2$. Emily C et al.'s study [49] using stair climbing as a form of high-intensity exercise, improved variations in $PeakVO_2$. Therefore, patients should choose appropriate training methods according to their own disease conditions and activities. This study also conducted subgroup analysis based on exercise frequency and duration but found no significant difference in intergroup comparison results, similar to the intergroup analysis presented by Goncalves et al [47] and Zheng et al [5] ($p = 0.79, I^2 = 0\%$) ($p = 0.25, I^2 = 24\%$). The reason for this could be that there is little difference between the results obtained among the subgroups, and only 2 studies were included for exercise frequency ($> 3$ times/week), while only 3 studies were included for exercise time ($> 40$ min). Therefore, any changes in these results should be interpreted carefully. In future research, more clinical studies are needed to verify the effects of HIIT and MCIT on $PeakVO_2$ considering different intervention timeframes, training modes, frequencies, and durations.

Heart rate serves as an indicator of myocardial oxygen demand, autonomic nerve regulation and balance, and is a crucial predictor of mortality in patients with cardiovascular diseases [45]. During exercise, the cardiovascular system adapts to meet the metabolic requirements of working muscles and thermoregulatory needs for skin blood flow while maintaining organ perfusion pressure. This adaptation leads to increased heart rate, cardiac output, and peak oxygen uptake through parasympathetic inhibition and sympathetic stimulation, ultimately

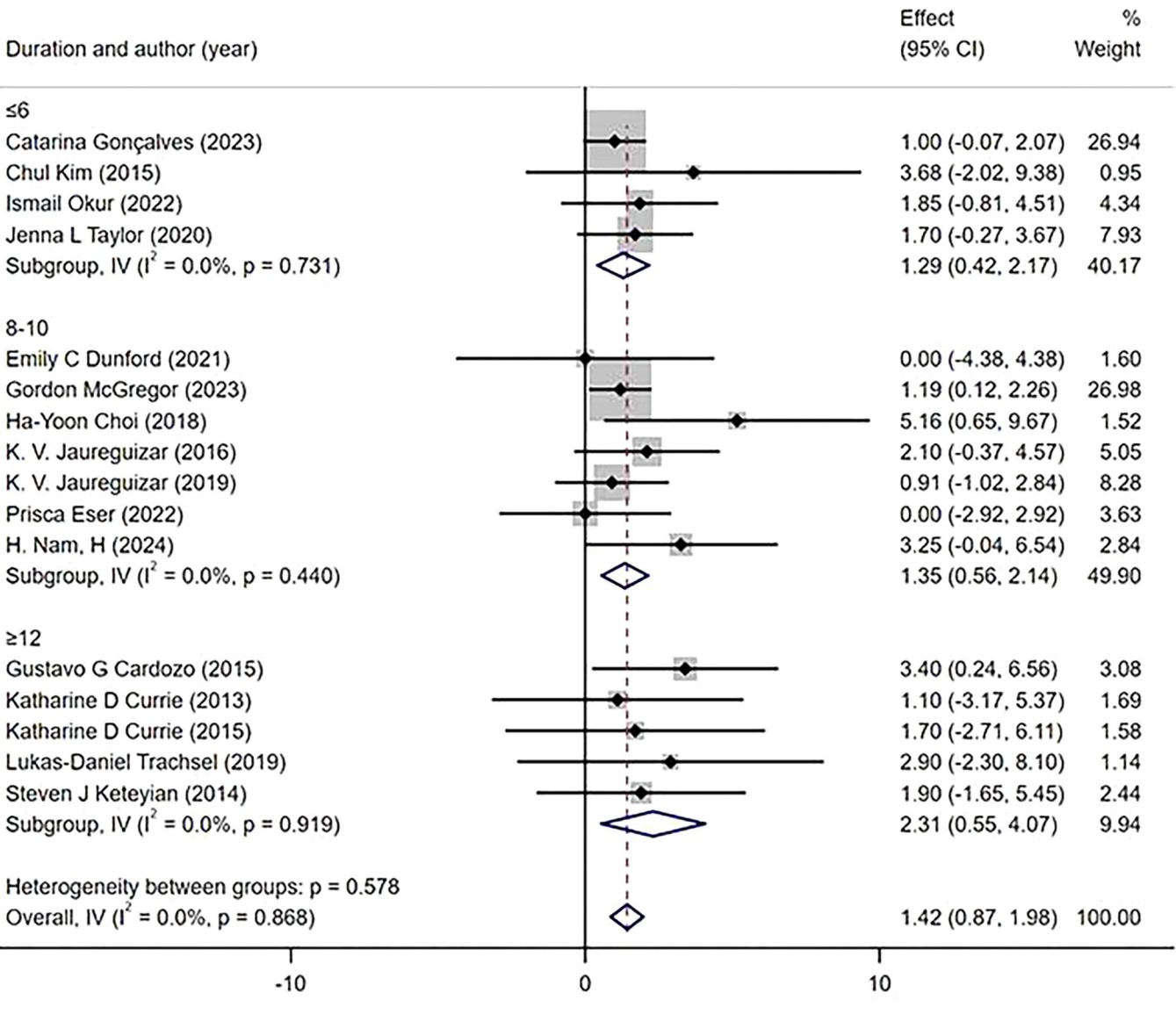

**Fig 10. Subgroup analysis of peak VO2.**

enhancing exercise performance [50]. In this study, we included 11 studies with PHR as the outcome measure. Our findings indicate that HIIT resulted in a greater increase in PHR compared to MICT, which aligns with Zheng et al.'s results [5], but differs from those reported by Qin et al. [51]. Qin systematic review did not reveal a significant difference in PHR between HIIT and control groups [MD = 0.74bpm; 95% CL (-2.82,4.30); P = 0.68]. The limited improvement effect observed may be attributed to Qin's inclusion of only six studies using PHR as an outcome measure with small sample sizes that lacked statistical power. Given that PHR is influenced by various factors such as age, gender, disease type, muscle mass, and daily activity capacity among patients; further exploration through detailed clinical studies involving larger samples is warranted [52].

The 6-minute walk test (6MWT) is utilized to assess the exercise capacity of patients undergoing cardiac rehabilitation. This experiment does not impose any weight load or resistance

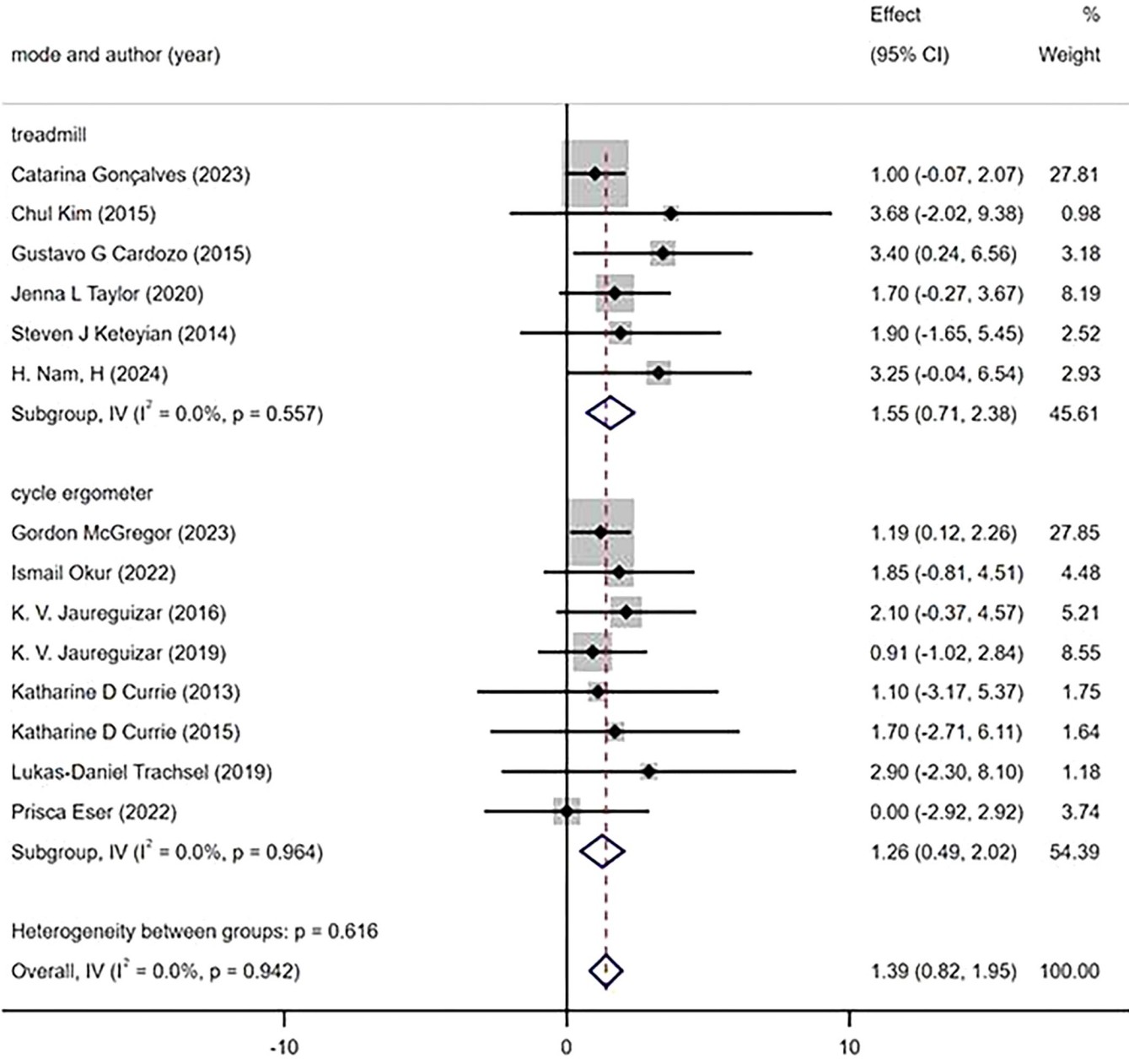

**Fig 11. Subgroup analysis of different exercise mode.**

on patients and evaluates their walking distance over a period of 6 minutes, thereby reflecting the outcomes following different training intensities [53]. The findings from this study revealed that patients receiving HIIT exhibited greater distances covered during the 6-minute walk compared to those undergoing MICT. A systematic review conducted by R Nicole, encompassing 15 studies on outpatient cardiac rehabilitation, demonstrated improvements in the 6MWT among rehabilitated patients [54], which aligns with our study's results. Maryam's investigation indicated that the maximum heart rate achieved during the 6MWT for patients engaged in cardiac rehabilitation exercises corresponded to approximately 78% of their maximum heart rate during cardiopulmonary exercise testing, thus contributing to enhancing their

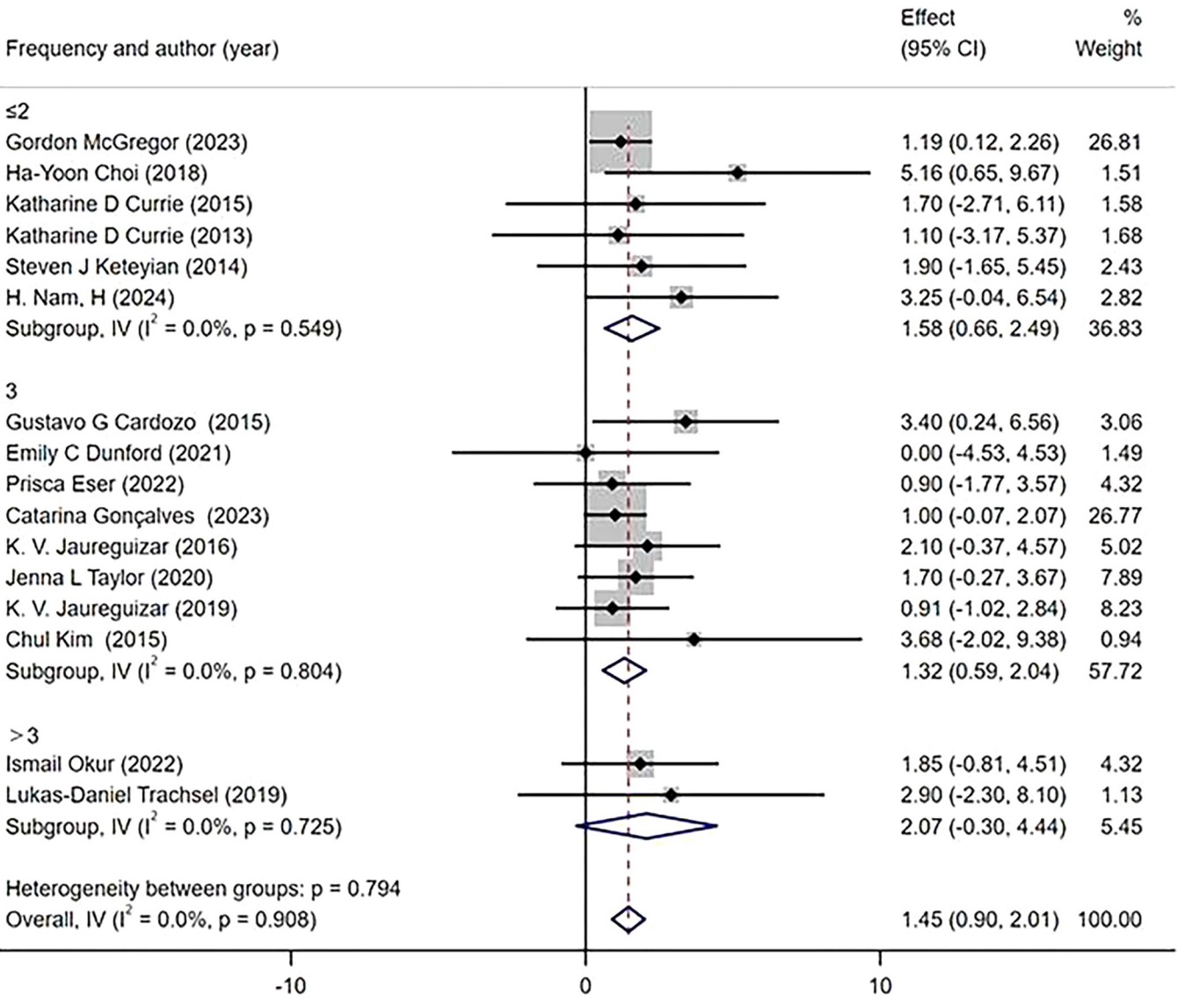

**Fig 12. Subgroup analysis of different exercise frequency.**

cardiopulmonary fitness levels. Furthermore, due to its ease of execution, this experiment can serve as an outcome measure for evaluating the exercise plan's intensity level among patients [50]. In our study, we included seven investigations employing the 6MWT as an outcome indicator; however, given the limited number of studies available, further verification is required regarding the increased walking distance observed in HIIT compared to MICT recipients. Consequently, future research should consider incorporating the use of 6MWT as an outcome measure when assessing various exercise programs' intensity levels among patients.

LVEF is a critical index for evaluating cardiac function in clinical practice, and an increase in LVEF levels can indicate improved cardiac function. The mechanism underlying the elevation of LVEF due to exercise intensity may be attributed to the reduction of left ventricular end-diastolic volume and end-systolic volume through high-intensity exercise, thereby enhancing ventricular remodeling and myocardial contractility [55]. This study included a

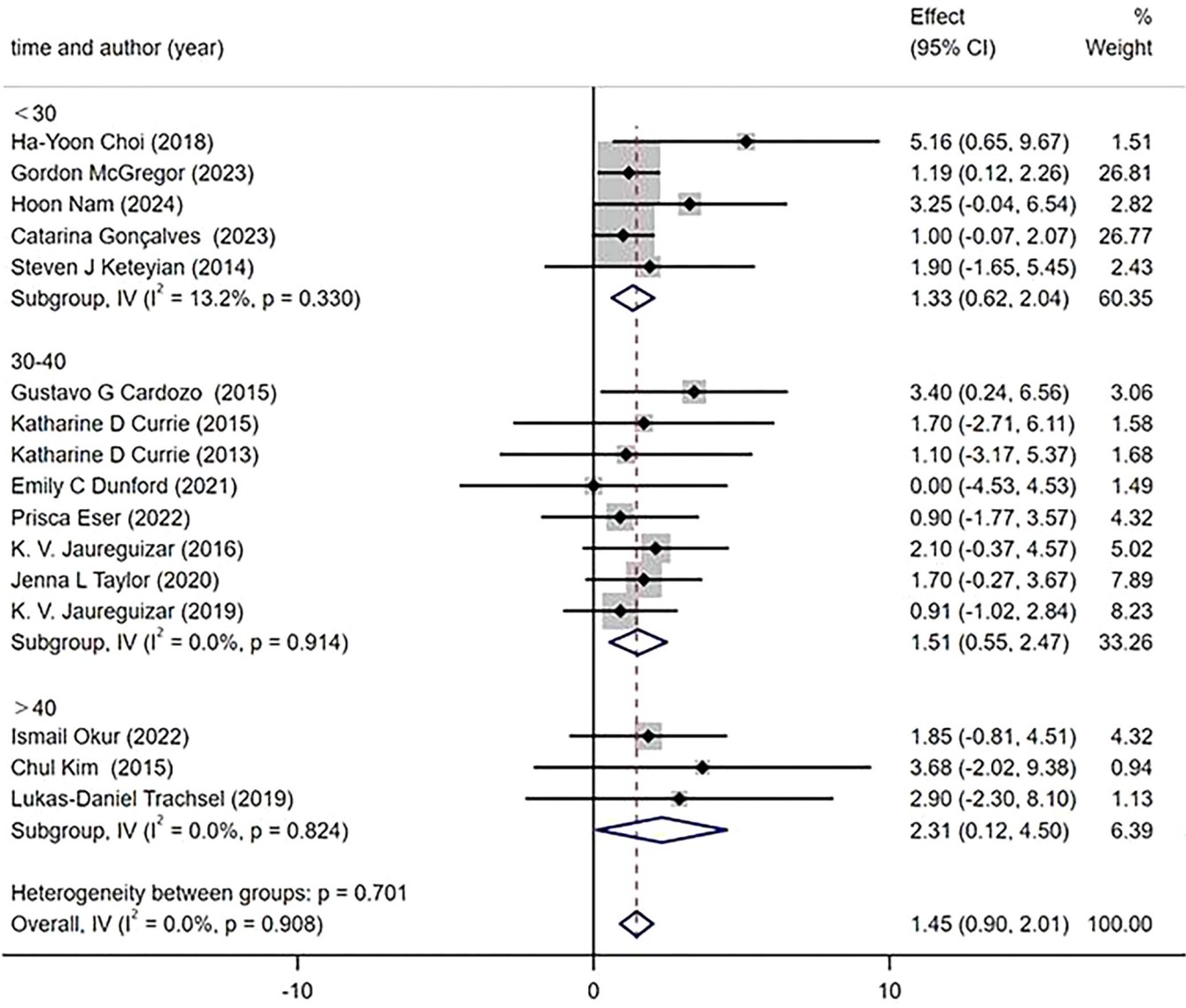

**Fig 13. Subgroup analysis of different exercise time.**

total of 7 literature sources with LVEF and LVEDV as outcome indicators; however, the sample size was inadequate. The results revealed no significant difference in the effectiveness of HIIT and MICT on improving LVEF and LVEDV, which aligns with Du's findings [45]. A large-scale study conducted by Øyvind et al. [56] demonstrated that neither HIIT nor MICT significantly improved LVEF in heart failure patients after 12 weeks. Therefore, more high-quality clinical studies are needed to support the improvement effects and mechanisms of different intensity exercises on cardiac function among cardiovascular patients.

A systematic review on blood pressure reduction for cardiovascular disease prevention reported that a decrease in systolic blood pressure by 10mmHg was associated with a 20% lower risk of cardiovascular events and a 13% decrease in all-cause mortality [57]. The findings of this study suggest no significant difference between HIIT and MICT regarding their effects on improving SBP. However, HIIT demonstrated superior effectiveness in improving DBP

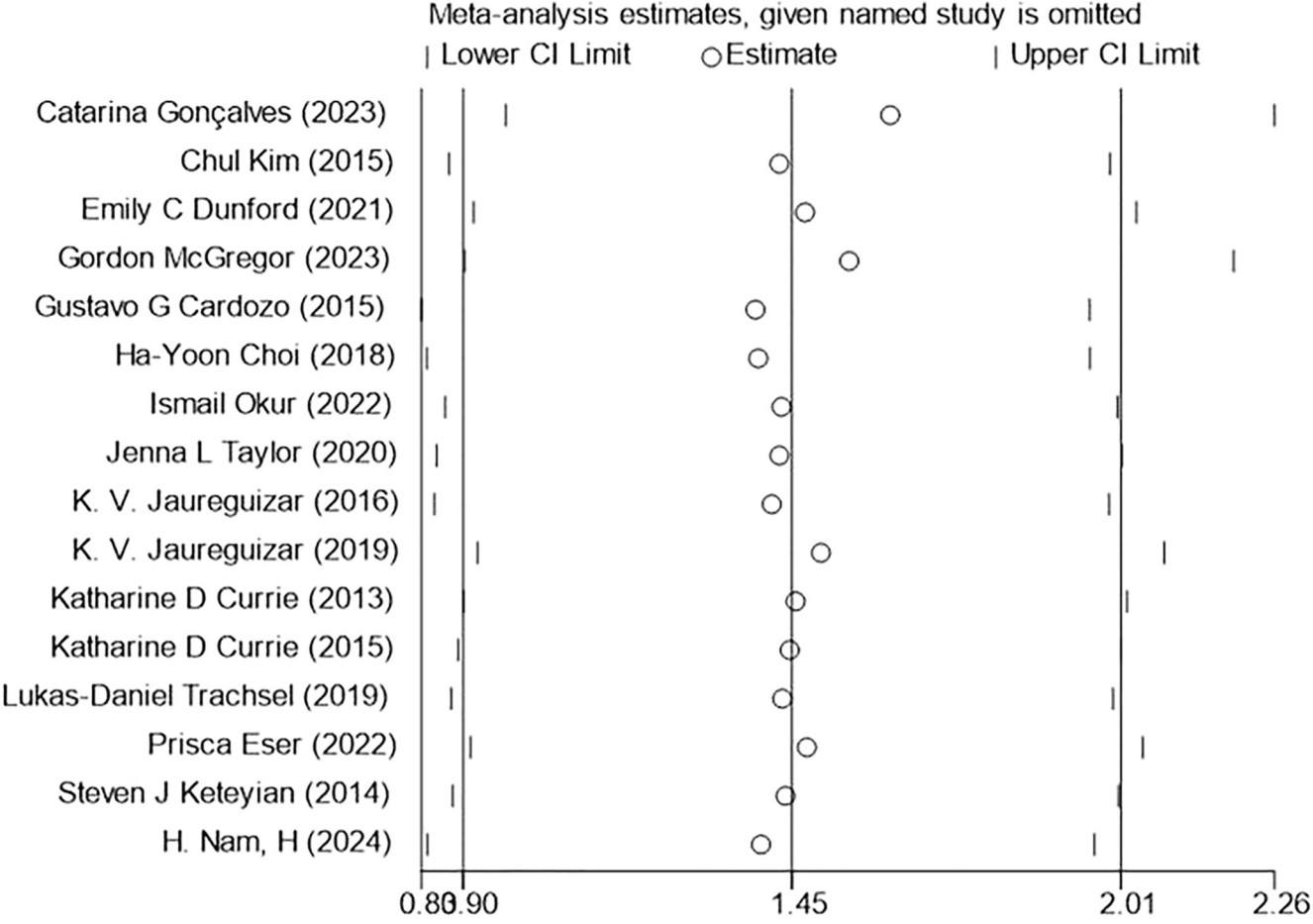

**Fig 14. PeakVO2 sensitivity analysis.**

compared to MICT. Moreover, the impact of exercise intensity on blood pressure improvement remains inconclusive, consistent with other studies [45,58], Du et al. [45] Furthermore, MICT appears to yield greater reductions in both systolic and diastolic blood pressure than HIIT.

A study investigating the impact of exercise on hypertensive patients revealed that individuals with hypertension exhibit vascular endothelial dysfunction and experience a wide range of blood pressure fluctuations [59]. Moderate intensity continuous training has been shown to enhance maximum oxygen intake and improve vascular endothelial function, thereby facilitating post-exercise reduction in blood pressure [60]. The limited effect observed across different exercise intensities on blood pressure improvement in this study may be attributed to the majority of included patients not exceeding the classification for hypertension, with only 3 studies reporting abnormal blood pressure levels [21,27,33]. Moreover, there was no significant improvement in blood pressure before and after intervention. Additionally, among the included literature, 18 studies reported baseline drug usage including beta-blockers, calcium channel blockers, and angiotensin-converting enzyme inhibitors. While two studies by Katharine D Currie [27,39] mentioned changes in patients' medication use, four studies [18,31–33] did not provide any information regarding drug usage. Therefore, further investigation is warranted to determine whether alterations in blood pressure regulated by varying

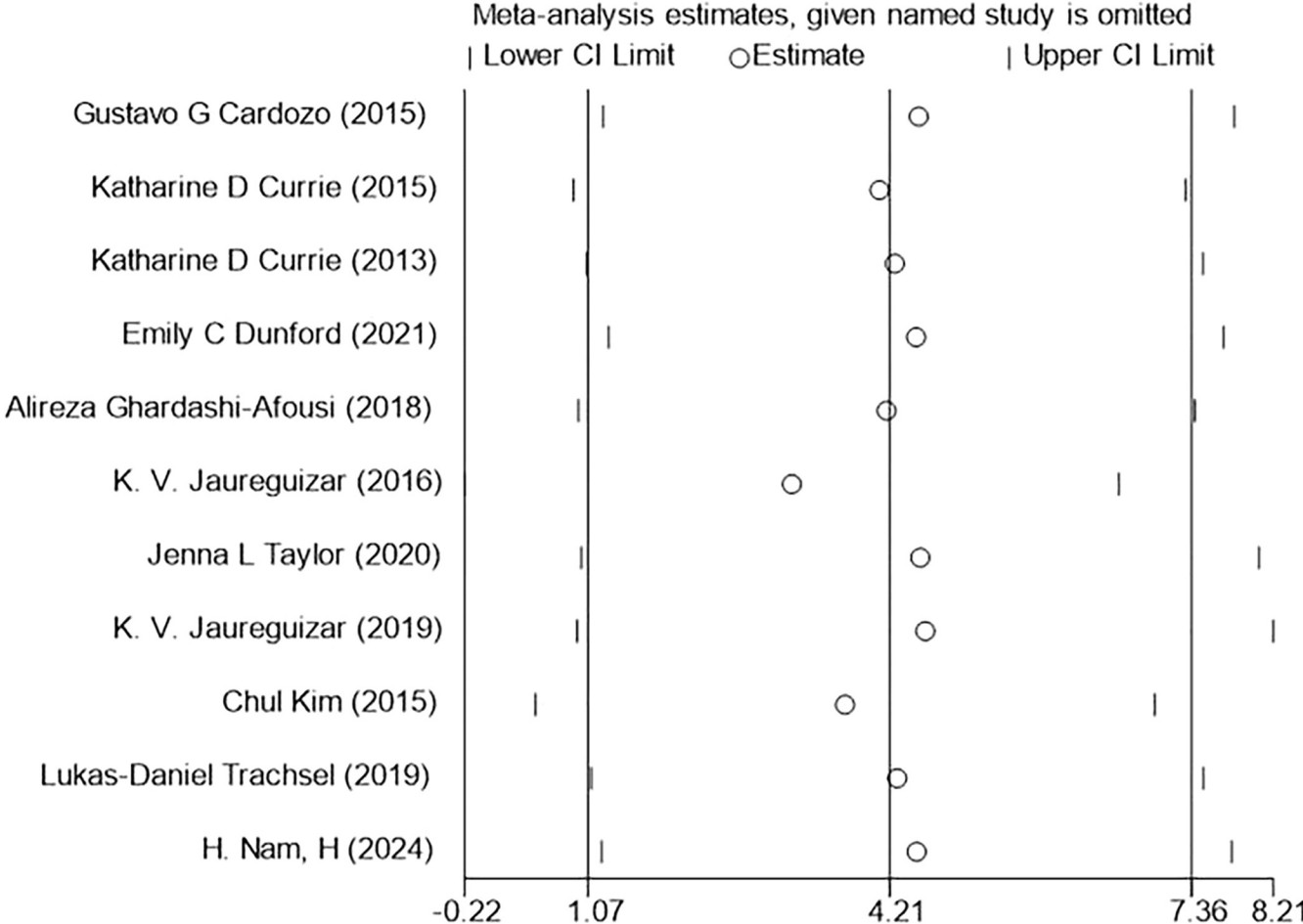

**Fig 15. PHR sensitivity analysis.**

intensity exercises are influenced by medication effects. Based on the comprehensive literature review conducted and the findings obtained from this study, it is reasonable to conclude that MICT surpasses HIIT when it comes to reducing blood pressure levels. This also implies that patients should select appropriate exercise modalities based on their individual conditions for maintaining balanced blood pressure.

## 5 Conclusion

The results of this systematic review showed that compared to MICT, HIIT had a greater improvement on PeakVO$_2$ among CAD patients. Furthermore, HIIT seemed unaffected by intervention duration, exercise mode, frequency or exercise session when it came to enhancing PeakVO$_2$. In addition, HIIT outperformed MICT when it came to improving 6MWT, PHR, and SBP. On the other hand, MICT proved more effective than HIIT at reducing DBP. Nevertheless, there were no significant differences observed between the effects of HIIT and MICT on SBP, LVEFand LVEDV.Moving forward, we hope for an increase in high-quality clinical controlled studies with larger sample sizes over longer periods of time so as to better evaluate how both forms of training affect CAD patients' cardiopulmonary levels and exercise ability.

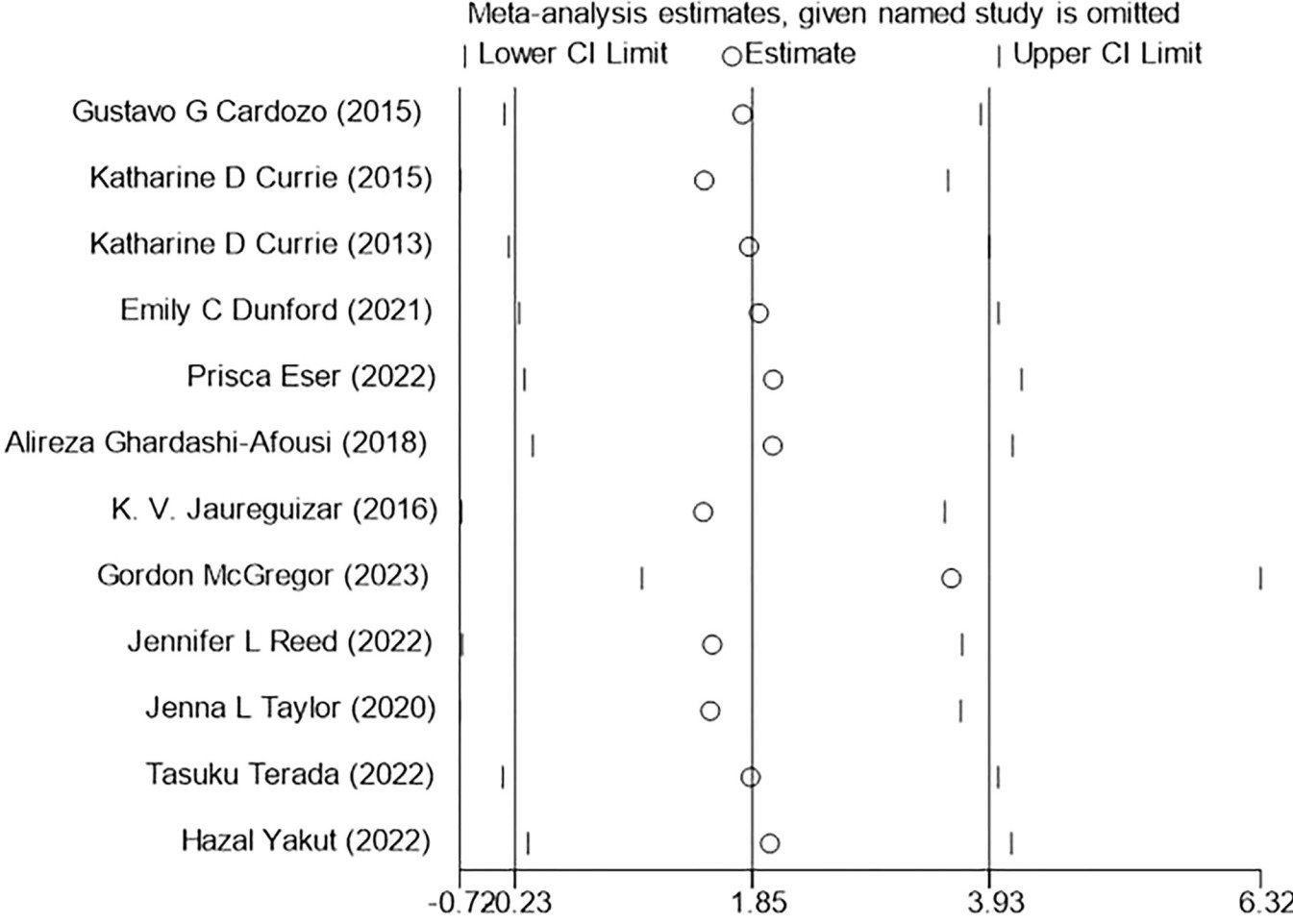

**Fig 16. SBP sensitivity analysis.**

## 6 Limitations

Limitations: (1) In this study, 15 participants mentioned that HIIT and MICT were completed under the supervision of researchers, while 7 did not mention whether the training plan was supervised. This lack of uniformity in exercise intensity and completion rate could not be completely addressed. (2) Most of the studies included in this analysis were small sample randomized controlled trials, with only one large sample study. Additionally, there was a predominance of male patients and a lack of female patients, resulting in potential heterogeneity due to gender differences in intervention content, intensity, and frequency. (3) Some studies did not utilize maximum heart rate as a measure of intervention intensity, and certain HIIT interventions did not reach 80%$HR_{max}$. These factors may introduce bias towards MICT when assessing intervention effects. (4) Nine specific randomization methods were identified among the included studies; seven specified the use of blinding; four specified apportionment concealment. These methodological variations increase the risk of result heterogeneity. (5) Due to differences in calculation methods for exercise intensity (such as $HR_{max}$, $HR_{peak}$, $VO_{2max}$, and Workload), along with limited inclusion of studies using certain calculation methods, result analysis may exhibit heterogeneity; therefore subgroup analysis was not conducted in this study.

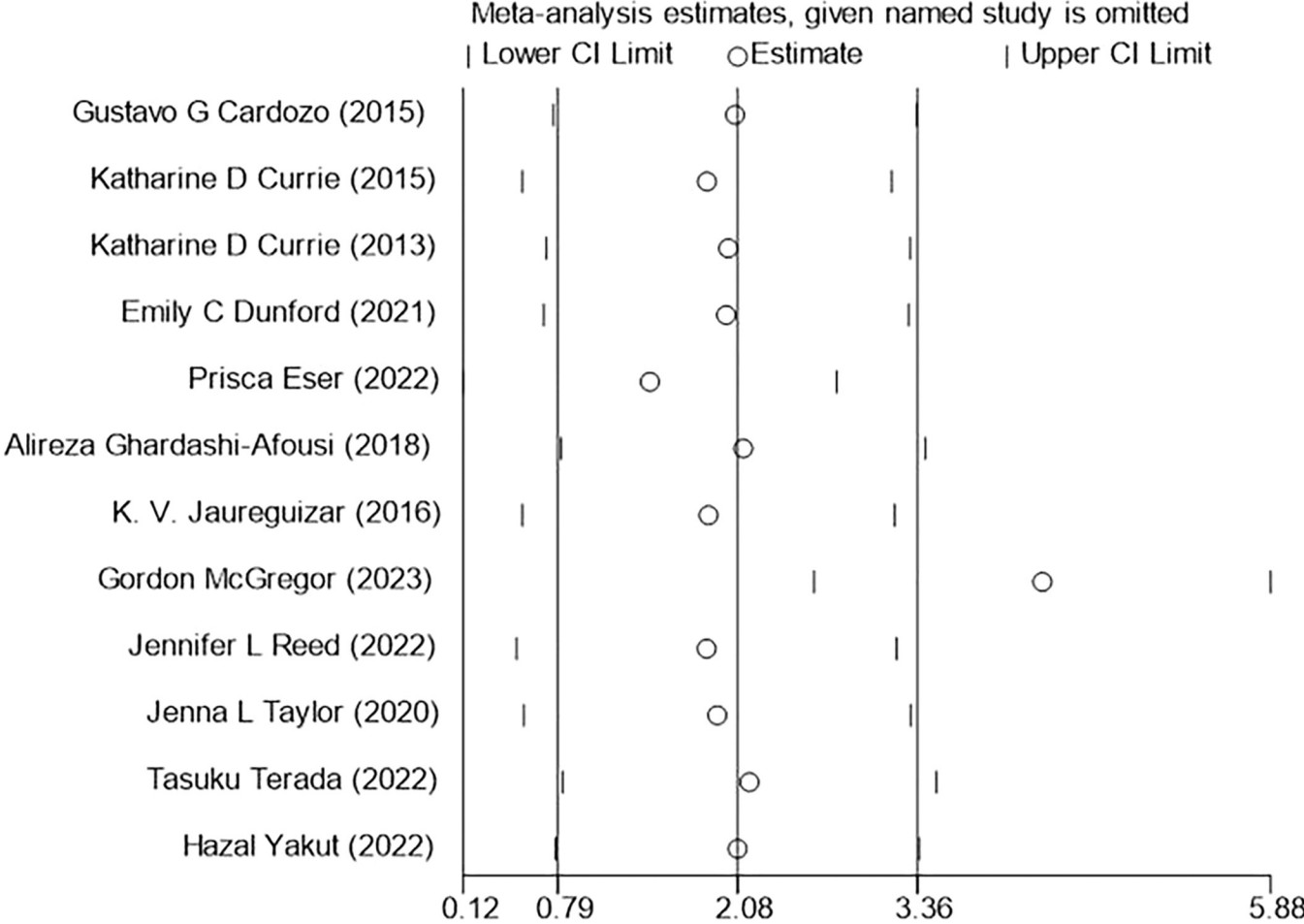

**Fig 17. DBP sensitivity analysis.**

## Supporting information

**S1 File. PRISMA 2020 checklist.**
(DOCX)

**S2 File. Search record.**
(DOCX)

**S3 File. GRADE of evidence.**
(DOCX)

**S4 File. Funnel plot and Egger's test.**
(DOCX)

**S5 File. ROB 2 risk assessment details.**
(XLSM)

## Acknowledgments

We thank all the authors for their tireless assistance in this project. We sincerely appreciate the reviewers' strict standards for quality and their valuable suggestions for further enhancements.

## Author Contributions

**Conceptualization:** Chao Gao.

**Data curation:** Chao Gao, Shuyao Zhu.

**Formal analysis:** Chao Gao, Junming Zhang.

**Funding acquisition:** Yuchuan Yue, Dongmei Wu.

**Validation:** Chao Gao, Junming Zhang, Shuyao Zhu.

**Visualization:** Chao Gao.

**Writing – original draft:** Chao Gao, Yuchuan Yue, Dongmei Wu.

**Writing – review & editing:** Chao Gao, Yuchuan Yue, Dongmei Wu.

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
