## [Decision Letter · Decision Letter 0]

24 Sep 2024

PONE-D-24-35383Effects of high-intensity interval training versus moderate-intensity continuous training on cardiorespiratory and exercise capacity in patients with coronary artery disease: a systematic review and meta-analysisPLOS ONE

Dear Dr. Yue,

Thank you for submitting your manuscript to PLOS ONE. After careful consideration, we feel that it has merit but does not fully meet PLOS ONE’s publication criteria as it currently stands. Therefore, we invite you to submit a revised version of the manuscript that addresses the points raised during the review process.

We look forward to receiving your revised manuscript.

Kind regards,

Prof. Jeremy B. Coquart, Ph.D.

Academic Editor

PLOS ONE

Journal Requirements:

"Project number: XJLL2023007;

 Funded by: The Fourth People's Hospital of Chengdu;

 Sponsor: Dongmei Wu(Writing – original draft);Yuchuan Yue (Preparation of the manuscript)"

3. As required by our policy on Data Availability, please ensure your manuscript or supplementary information includes the following: 

Reviewers' comments:

Reviewer's Responses to Questions

**Comments to the Author**

1. Is the manuscript technically sound, and do the data support the conclusions?

Reviewer #1: Yes

Reviewer #2: No

2. Has the statistical analysis been performed appropriately and rigorously? 

Reviewer #1: Yes

Reviewer #2: Yes

3. Have the authors made all data underlying the findings in their manuscript fully available?

Reviewer #1: Yes

Reviewer #2: No

4. Is the manuscript presented in an intelligible fashion and written in standard English?

Reviewer #1: Yes

Reviewer #2: Yes

5. Review Comments to the Author

Reviewer #1: Hello,

Thanks for submitting your manuscript titled Effects of high-intensity interval training versus moderate-intensity continuous training on cardiorespiratory and exercise capacity in patients with coronary artery disease: a systematic review and meta-analysis

to this journal. It is a well written manuscript however I have following observations and comments

1. The abstract mentions significantly better outcomes with treadmill as compared to cycle ergometry but the methods or results does not mention the parameters based on which the same was concluded. Hence it would be advisable either to remove this in abstract or substantiate it with the data in the results

2. P value for various parameters mentioned significant in comparison is mentioned >0.05 and though we understand it is the p value for interaction and not for test of significance, it needs to be mentioned explicitly how the significance was concluded as mentioning P value >0.05 in the same line without clarifying on the same makes it confusing.

3. Please review and add flowchart of methods of meta analysis

Thanks

Reviewer #2: Define PeakVO2 in the keywords rather than using an abbreviation.

In the ‘results’ section of the abstract, most of the abbreviations are not defined. This part is difficult to understand/read.

The corresponding author should use a professional email address: 1903222804@qq.com

The first paragraph of the introduction is long and not interesting because it is very general. I would recommend getting to physical activity more quickly.

In the introduction, the authors justify their study by explaining that the literature on the topic lacks precision on the duration of programmes, but to compare the 2 exercise modalities the most important thing is not the duration of the programmes (4 vs 16 weeks), but also the intensity and duration and frequency of session. MICT can be at 50, 60, 70% VO2peak and, depending on the studies, HIIT can also vary from 40 to 110% VO2peak. Similarly, the duration of the session (28 min vs 40 min) or the frequency (1 vs 5 session/week) is also important... Without these analyses, the conclusions are likely to be erroneous (see table 2).

In the method, it is surprising to see that 1668 were identified in June, and that the paper was submitted at the beginning of September, just three months later... with 153 which were evaluated for full-text reading.

The authors should clearly indicate how the Boolean operators were used to combine the keywords/concepts. It would be interesting to clearly indicate the ‘phrase’ used/inserted (in PubMed…), so that we can check. At the moment, I'm finding that the number of articles is very different from the number mentioned by the authors.

In my opinion, in general, the method needs to be developed. Although I am familiar with the PRISMA standards, it is not currently possible to replicate this meta-analysis. Each stage of procedure must be detailed.

In the results section, there are far too many figures. The article contains 10 figures, some of which include 4 sub-figures.

In the discussion, the authors should take into account the heterogeneity of the studies, and therefore be more cautious.

6. PLOS authors have the option to publish the peer review history of their article (what does this mean?). If published, this will include your full peer review and any attached files.

Reviewer #1: **Yes: **Dr Kamal Sharma

Reviewer #2: No

---

## [Author Response · Author response to Decision Letter 0]

8 Oct 2024

Response Letter

Dear editors and reviewers,

 We are very grateful for your constructive comments and suggestions for our manuscripts entitled “Effects of high-intensity interval training versus moderate-intensity continuous training on cardiorespiratory and exercise capacity in patients with coronary artery disease: a systematic review and meta-analysis”(ID:PONE-D-24-35383).Your comments are very valuable and helpful for improving our manuscripts.in the following,the responses to all the comment are provided one by one.

 We have tried our best to make all the revisions clear,and we hope that the revised manuscript can satisfy the requirements for publication.

 The main revisions in the new manuscript are:

Reply to the editors suggestions for article modification

1.This study was modified according to the format template for viewing PLOS ONE, which meets the format requirements of PLOS ONE.

2.We supplemented our research with the help of funders: We are very grateful to the Chengdu Fourth People's Hospital for supporting our work, especially funders Wu Dongmei and Yue Yuchuan for their help in research design, financial support and manuscript preparation.

We have added the relevant information of Financial Disclosure: 

Project number: XJLL2023007; 

Funded by: The Fourth People's Hospital of Chengdu; 

Funder: Dong mei Wu(Funders of this study);Yue Yuchuan (Funders of this study).

3.We have described in the manuscript a table of the authors, years and other necessary contents of the included documents in this study, as well as the reasons for the excluded documents. All the included and excluded documents do not include unpublished studies.

4.We have uploaded a risk and quality assessment table of literature bias conducted in this study, which includes the reasons for inclusion, in a supplementary document3,5.

Reply to the reviewer's suggestions for article modification

Reply to the reviewer 1

1.The content of treadmill and Cycle ergometer you mentioned has been deleted in the abstract.See page 2-3 for details.

2.According to your suggestion that the P-value of various parameters of significance is >0.05, we modified the corresponding expression in the article by consulting the relevant system evaluation of HIIT and MICT and based on the data analysis results of this study.Has been marked in the manuscript.

3.About the content of the flow chart you mentioned. We have revised and updated the flowchart content.See Fig 1 for details.

Reply to the reviewer 2

1.According to your definition of PeakVO2 in keywords, we have provided the full name in the keyword section.

2.Regarding the email address(1903222804@qq.com) you mentioned, I have consulted the corresponding author, and this is the only email address she used in the submission process.

3.In the preface you mentioned, we have simplified the epidemiology of cardiovascular diseases in the preface, in order to highlight the benefits of exercise on exercise ability of cardiovascular diseases and introduce the focus of this study.See page 3-5 for details.

4.According to your suggestion that HIIT and MICT should take into account the effects of different exercise intensity, duration, and frequency on PeakVO2, we sincerely appreciate your input. In response, we have extended the duration of each exercise session and increased the weekly frequency of both HIIT and MICT for subgroup analysis. You can find detailed information in the Fig8(c d) provided on the article page. However, due to variations in calculation/evaluation methods used for exercises with different intensities (such as HRmax, HRpeak, VO2max, and Workload), this study did not conduct subgroup analysis based on exercise intensity. Additionally, some evaluation methods had a varying number of indicators. For instance, only one study utilized VO2max as a measure of training intensity. Combining or conducting subgroup analysis using different calculation/evaluation methods could introduce heterogeneity in results or bias towards subgroups with more literature available. As we did not perform this specific subgroup analysis mentioned above, we have acknowledged this limitation in our explanation. 

5.According to your PRISMA standard suggestions, we have added the specific details of each process in PRISMA, such as adding the location of images and supplementary files.

6.According to your suggestion, by referring to similar systematic evaluation and combining with the analysis of the results of this study, we have decided to include the pictures with publication bias in the attachment, and this decision will not increase the difficulty of understanding the full text.

7.According to your suggestions, this study has re-searched the following databases (Pubmed, Web of Science, Embase, Cochrane Library, CLNAHL) using the specified search terms. Based on the input from researchers Gao and Zhang, we included another researcher Zhu to collaborate in literature screening. During the screening process, we encountered a disagreement and sought advice from a fourth researcher Yue. Throughout the screening process, we retained and labeled documents that were included in the initial search while excluding others. We also compared documents from the first search with those from the second search. As of September 26th, a total of 1432 literatures were searched; however, no updated literature relevant to this systematic evaluation was found. The flow chart has been updated accordingly and Boolean operation symbols have been added to our search files. No filters were applied during retrieval.

8.Based on your suggestions, we added a more cautious interpretation of the results and discussion by referring to other studies and reviewing the discussion.

Sincerely,

Corresponding author.

---

## [Decision Letter · Decision Letter 1]

6 Nov 2024

Effects of high-intensity interval training versus moderate-intensity continuous training on cardiorespiratory and exercise capacity in patients with coronary artery disease: a systematic review and meta-analysis

PONE-D-24-35383R1

Dear Dr. Yue,

We’re pleased to inform you that your manuscript has been judged scientifically suitable for publication and will be formally accepted for publication once it meets all outstanding technical requirements.

Kind regards,

Jeremy B. Coquart, Ph.D.

Academic Editor

PLOS ONE

Additional Editor Comments (optional):

Reviewers' comments:

Reviewer's Responses to Questions

**Comments to the Author**

1. If the authors have adequately addressed your comments raised in a previous round of review and you feel that this manuscript is now acceptable for publication, you may indicate that here to bypass the “Comments to the Author” section, enter your conflict of interest statement in the “Confidential to Editor” section, and submit your "Accept" recommendation.

Reviewer #1: All comments have been addressed

2. Is the manuscript technically sound, and do the data support the conclusions?

Reviewer #1: Yes

3. Has the statistical analysis been performed appropriately and rigorously? 

Reviewer #1: Yes

4. Have the authors made all data underlying the findings in their manuscript fully available?

Reviewer #1: Yes

5. Is the manuscript presented in an intelligible fashion and written in standard English?

Reviewer #1: Yes

6. Review Comments to the Author

Reviewer #1: Hello,

The authors have addressed , revised and corrected the manuscript as per the review submitted. The manuscript is now improved and can be accepted in the revised format for publication.

Thanks

7. PLOS authors have the option to publish the peer review history of their article (what does this mean?). If published, this will include your full peer review and any attached files.

Reviewer #1: **Yes: **Dr Kamal H sharma

---

## [Editor Report · Acceptance letter]

4 Dec 2024

PONE-D-24-35383R1 

PLOS ONE

Dear Dr. Yue, 

I'm pleased to inform you that your manuscript has been deemed suitable for publication in PLOS ONE. Congratulations! Your manuscript is now being handed over to our production team.

Kind regards, 

on behalf of

Professor Jeremy B. Coquart 

Academic Editor

PLOS ONE